# Specific and Intraspecific P Efficiency of Small-Grain Legumes as Affected by Long-Term P Management

Yue Hu [1,2], Klaus J. Dehmer [1], Evelin Willner [1] and Bettina Eichler-Löbermann [2,*]

[1] Satellite Collections North, Genebank, Leibniz Institute of Plant Genetics and Crop Plant Research (IPK), 23999 Malchow, Germany
[2] Agronomy and Crop Science, University of Rostock, 18059 Rostock, Germany
* Correspondence: bettina.eichler@uni-rostock.de

**Abstract:** Legumes have a high demand for phosphorus (P) but also have effective physiological and morphological strategies of P mobilisation. In order to evaluate the inter- and intraspecific P efficiency of small-grain legumes under contrasting long-term P management, eight accessions each of alfalfa (*Medicago sativa* L.) and red clover (*Trifolium pratense* L.) were cultivated in two consecutive growing periods from 2020 to 2021 in a field trial established in 1998. Six treatments (no P, triple-superphosphate (TSP), biomass ash, cattle manure, biowaste compost, and biowaste compost + TSP) were considered as P sources. While the yield clearly varied between both growing seasons, the differences between alfalfa and red clover were relatively small (4.7 vs. 4.9 Mg ha$^{-1}$ in 2020 and 12.0 vs. 10.5 Mg ha$^{-1}$ in 2021, $p < 0.05$). Even after more than 20 years of P management, crop yields were hardly affected by mineral P sources (TSP and biomass ash) while organic fertilisers increased the yields and nutrient uptake of plants and also raised soil P pools and the activities of soil enzymes in comparison to the control. A relevant crop effect was only found for the nitrogen (N) leaching with higher mineral N contents in 60 to 90 cm soil depth measured for red clover compared to alfalfa (11.8 vs. 4.8 kg ha$^{-1}$, $p < 0.05$). Our results emphasise the high P efficiency of small-grain legumes without pronounced inter- or intraspecific differences. The yield-enhancing effect of organic amendments was related to higher soil fertility rather than to P supply.

**Keywords:** organic amendments; alfalfa; red clover; phosphorus utilisation





## 1. Introduction

Alfalfa (*Medicago sativa* L.) and red clover (*Trifolium pratense* L.) are perennial small-grain legumes with great importance as forage and protein-rich feeds worldwide [1–4]. Due to the biological nitrogen (N) fixation (BNF) through the symbiotic association of rhizobia, these crops can contribute to economic value and sustainability in plant production [5,6]. The N input from BNF by alfalfa and red clover can be more than 300 kg N ha$^{-1}$ year$^{-1}$ [7,8] and serves as a sustainable source of N to replace chemical fertiliser inputs [9,10]. Numerous studies have proven the benefits of alfalfa and red clover in cereal- or maize-based crop rotations [11–13]. Further beneficial traits of these legumes include good persistence and wide adaptability to various conditions [14,15].

However, the high N contribution caused by BNF may also increase the concentration of mineral N in soil and consequently the risk of nitrate (NO$_3$-N) leaching after plant growth [16]. After the cultivation of alfalfa and red clover, mineral N up to about 250 kg N ha$^{-1}$ can be released [17,18]. These N inputs by legumes urgently need to be considered in fertiliser planning to avoid the accumulation of excessive NO$_3$-N in soil [19].

Phosphorus (P) plays a crucial role during BNF, and plants engaged in BNF generally have a high P requirement [20]. The BNF is an energetically expensive process, consuming 16 adenosine triphosphate (ATP) molecules for producing two NH$_3$ molecules [21]. Additionally, P is a structural component of nucleic acids, coenzymes, and phospholipids

and may be a critical constraint for legumes in low-nutrient environments [22]. The high P demand of legumes increases their sensitivity to P deficiency, which can be a major limiting factor for legume production [23–25].

In comparison to the conventional chemical P fertilisers such as triple-superphosphate (TSP), recycling fertilisers such as biomass ash, biowaste compost, and cattle manure have been proven to be an adequate P source [26–29]. Besides the pure P effect, many recycling fertilisers have further effects on soil quality, e.g., pH value, organic matter content, and microbial activity, which finally can also affect plant growth and development [30–32]. Previous experiments, including the experimental site of this study, have shown that the responsiveness of crops to P application differs. While for maize or potatoes clear P fertiliser effects were usually found, winter cereals rarely had higher yields after P application [33,34].

To cope with the limitation under low P availability, legumes are usually efficient in P acquisition and mobilisation from the soil. Morphologically, P deficiency can induce alterations in lateral root development and architecture, root hair formation, and cluster root development [35–37]. Physiologically, legume plants can increase the synthesis and exudation of low-molecular-weight organic acids [38–40] and mobilise sparingly available P via chelation as well as ligand exchange [41]. In addition, phosphatases can be released under P deficiency to hydrolyse and utilise organic P compounds in soil. While plants mainly release acid phosphatase (acP), alkaline phosphatase (alP) is released mainly by bacteria, fungi, and earthworms [42].

Several studies have confirmed interspecific differences for soil P acquisition between legume species, reported mostly for variations regarding root traits and root exudations [40,43–46]. Phosphorus acquisition strategies were also described for *Trifolium* and *Medicago* species [46–49]. However, a direct comparison of P uptake and efficiency between alfalfa and red clover is to our knowledge not yet available. Additionally, intraspecific variation in P acquisition has been studied within different legume species such as white clover [50], subterranean clover [51], common bean [52], chickpea [53,54], and faba bean [55]. Genotypic studies of alfalfa and red clover have been mainly concentrated on forage and seed yield, nutritive value, and tolerance to stress [56–59], whereas P uptake and utilisation studies between alfalfa and red clover genotypes under different P supplies are scarce. Nevertheless, based on the studies regarding the intraspecific P utilisation in other legume species, one can also expect genotypic variations in P acquisition in alfalfa and red clover genotypes.

To investigate the inter- and intraspecific P efficiency of alfalfa and red clover as affected by mineral and organic fertilisers, we conducted a two-year consecutive field trial with eight accessions each of alfalfa and red clover, aiming to (1) evaluate the P efficiency of different accessions of alfalfa and red clover, (2) explain crop-induced changes in soil characteristics, and (3) assess the ability of alfalfa and red clover to utilise P from different P sources. Thus, this investigation cannot only provide valuable information for the accession selection during the breeding process, but may also improve fertiliser management in legume forage production.

## 2. Material and Methods

### 2.1. Preliminary Experiment

In 2019, 149 alfalfa and 120 red clover accessions of the IPK Gene Bank (Satellite Collections North, Leibniz Institute of Plant Genetics and Crop Plant Research) were cultivated in experimental fields in the northeast of Germany. The accession selection was performed based on multiple parameters including geographic origin of the plant material, sample status, plant P concentration, and maturity group (Table 1). Eight accessions each of alfalfa and red clover were chosen as representatives. More information on the accessions studied is available on the homepage of [60].

**Table 1.** Geographic origin, sample status (SAMPSTAT), and plant P content of selected accessions of alfalfa and red clover.

| Alfalfa | | | | Red Clover | | | |
|---|---|---|---|---|---|---|---|
| Accession | Origin | SAMPSTAT | Plant P Concentration (mg kg$^{-1}$) | Accession | Origin | SAMPSTAT | Plant P Concentration (mg kg$^{-1}$) |
| LE2812 * | YEM | 300 | 4.29 | LE1731 * | KGZ | 300 | 3.95 |
| LE2368 | FRA | 500 | 4.13 | LE1423 | FIN | 400 | 3.67 |
| LE2370 | DNK | 500 | 3.94 | LE1391 | GBR | 200 | 3.56 |
| LE2521 | DEU | 500 | 3.80 | LE2750 | HRV | 100 | 3.43 |
| LE713 | ROU | 500 | 3.03 | LE1599 | DEU | 300 | 3.17 |
| LE888 * | DEU | 500 | 2.91 | LE1775 | RUS | 100 | 2.98 |
| LE2669 | ROU | 300 | 2.51 | LE1804 | SUN | 999 | 2.83 |
| LE2511 | FRA | 500 | 2.44 | LE1937 * | DEU | 100 | 2.72 |

Origin: country codes according to ISO 3166: SAMPSTAT: 100 = wild; 200 = weedy; 300 = traditional cultivar/landrace; 400 = breeding/research material; 500 = advanced or improved cultivar; 999: other. Accessions with asterisks were further selected for soil sampling in different depths.

*2.2. Experimental Design*

A consecutive field trial was conducted from 2020 to 2021 at the experimental station of the University of Rostock, based on a long-term field experiment established in 1998. The experimental site is located in a maritime-influenced region in northeast Germany (54°3′41.47″ N; 12°5′5.59″ E), about 15 km south of the Baltic Sea shore. The mean annual precipitation is around 600 mm and the average annual temperature is 8.1 °C. Precipitation and temperature during this study are shown in Figure 1. The soil at the study site is classified as Stagnic Cambisol according to the World Reference Base for Soil Resources [61], and the soil texture is loamy sand. In early March 2020, prior to sowing, the double lactate extractable P (Pdl), as an indicator of plant available P, ranged between 29.9 and 55.8 mg kg$^{-1}$ (Table 2). According to German soil P test classifications, these values can be considered to be low to sufficient considering 40 to 60 mg kg$^{-1}$ as the threshold to be achieved [62,63].

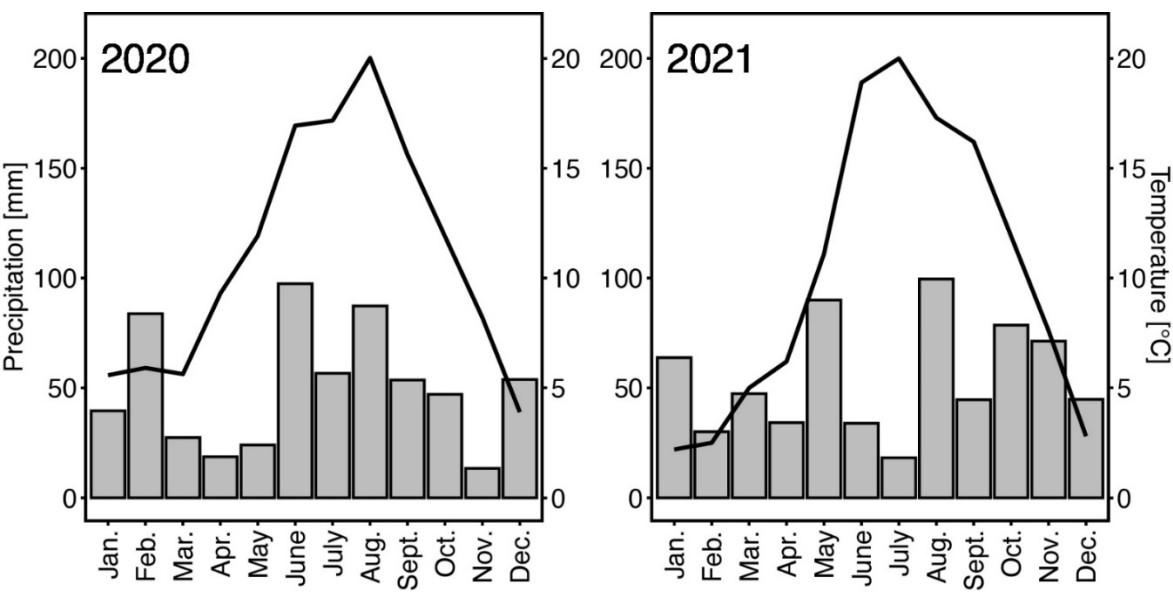

**Figure 1.** Average monthly precipitation (mm, shown as bars) and average monthly temperature (°C, shown as lines) in 2020 and 2021.

**Table 2.** Phosphorus (P) input, P offtake, and P budget from 1998 to 2020 and the double lactate extractable P (Pdl) measured in early March 2020 of the six studied treatments.

| Treatment | P Input | P Offtake | P Budget | P Input | P Offtake | P Budget | Pdl in March 2020 |
|---|---|---|---|---|---|---|---|
| | Cumulatively | | | Annually | | | |
| | kg ha$^{-1}$ | | | kg ha$^{-1}$yr$^{-1}$ | | | mg kg$^{-1}$ |
| No P | 0 | 524 | −524 | 0 | 23 | −23 | 29.9 |
| TSP | 559 | 584 | −25 | 24 | 25 | −1 | 44.7 |
| Ash | 438 | 553 | −115 | 19 | 24 | −5 | 42.5 |
| Manure | 510 | 590 | −79 | 22 | 26 | −3 | 43.5 |
| Compost | 611 | 587 | 24 | 27 | 26 | 1 | 44.0 |
| Compost + TSP | 1176 | 624 | 553 | 51 | 27 | 24 | 55.8 |

No P = without P supply, TSP = triple-super-phosphate, ash = biomass ash. Manure = cattle manure, compost = biowaste compost.

The field trial was arranged in a randomised split-plot design with four blocks (previously described by the authors of [34,64]). Each block consisted of three main plots (408 m$^2$ each) supplied with organic fertilisers. Within each main plot, mineral fertilisers were applied in three subplots (120 m$^2$ each), resulting in nine fertiliser combinations. Within each subplot, selected accessions of alfalfa and red clover were cultivated in sub-subplots with a size of 1.5 m × 2.5 m. From the nine fertiliser treatments, six were selected for this study: a treatment without P supply (no P), with triple-superphosphate (TSP), biomass ashes (ash, incinerated plant residues), cattle manure (manure), bio-waste compost (compost, sanitised compost based on green garden and landscape waste residues), and a combination of compost and TSP (compost + TSP). From 1998 to 2013, TSP was applied annually in autumn at a rate of 21.8 kg P ha$^{-1}$yr$^{-1}$ and increased to 30 kg P ha$^{-1}$yr$^{-1}$ in 2014 to balance the negative P budget. Ash treatment was first applied in 2007 with 63 kg P ha$^{-1}$yr$^{-1}$, and then with varying rates on the subsequent application dates. Manure and compost were applied triennially in autumn at a rate of around 30 Mg ha$^{-1}$ starting in 1998, with the exception being that the fertilisation in autumn 2019 was postponed to late March 2020 due to unfavourable weather conditions. The P input, offtake, and budget of the treatments from 1998 to 2020 are shown in Table 2. Nutrients other than P as well as the pH value in soil were balanced over the experimental period. The soil organic matter contents (SOM) were 2.52 and 2.30 for treatments with and without organic fertilisers in 2004 [64], and 2.64 and 2.27 in 2012 [29].

Based on the guidelines of the Federal Plant Variety Office [65], seeds of each accession equivalent to 1000 seeds m$^{-2}$ were sown in late April 2020. After seeding, rhizobia (RADICIN-Trifol, Jost, Germany) were applied for inoculation. Nitrogen was applied as calcium ammonium nitrate in 2020 in the amount of 30 kg ha$^{-1}$. Magnesium (Mg) and potassium (K) were applied in previous years and were in optimal ranges. Weeds were removed manually during the cultivation period. Plant biomass was harvested twice a year at the flowering stage, with aboveground biomass in each plot being cut about 7 cm above the soil, weighed, and sampled for laboratory analyses. In 2020, the first and second cuttings were in July and August, while in 2021, they took place in June and July. After the final harvest in 2020, plants remained in the field over winter for the next consecutive cultivation year.

Soil samples at a depth from 0 to 30 cm were taken for all accessions each year after the second harvest for standard soil P tests. Three accessions each of alfalfa and red clover with high, intermediate, and low plant biomass at the first harvest in 2020 were selected as candidates for the determination of soil enzyme activities, and the soil in these plots was sampled in 0 to 20 cm soil depths in October of each year. Two accessions each of alfalfa and red clover with the highest and lowest plant biomass at the first harvest in 2020 were chosen as candidates for Nmin and Pdl measurements in the soil profile, where soil

samples from 0 to 30, 30 to 60, and 60 to 90 cm were taken in October in 2020 and 2021. All soil samples were taken with an auger from two randomly selected spots in each plot and mixed carefully. Air-dried soil samples were sieved (2 mm) for standard soil P tests (Pdl). Fresh soil samples were taken and stored at −12 °C for laboratory determination of enzyme activities and Nmin.

### 2.3. Plant and Soil Analysis

Plant P concentration was determined after dry ashing and HCl digestion (2 g dry material digested in 20 mL of 25% boiling HCl in a 100 mL volumetric flask) via inductively coupled plasma optical emission spectroscopy (ICP-OES; ICP Serie Optima 8300DV, PerkinElmer Waltham, MA, USA). The total N concentration was determined using a CNS analyser (vario EL cube, elementar, Hanau, Germany). The N and P uptake by the plant were determined as the product of the plant N and P concentration and the dry weight of harvested biomass, and were given as a sum of the first and second cuttings. The N uptake was only measured for 2021. The N:P ratio in plant biomass was calculated using the ratio between N and P concentration.

For the standard soil P test, the Pdl content was determined using a modified method of [66]. Briefly, 12 g air-dried and sieved soil (2 mm) were mixed with a 150 mL solution consisting of calcium lactate (0.4 M $C_6H_{10}CaO_6 \times 5H_2O$) and hydrochloric acid (0.5 M HCl) at pH 3.6 and were shaken overhead for 90 min with 35 rpm. After filtration (cellulose round filter, 3 to 5 μm), P concentration in the filtrate was determined via ICP-OES.

The mineral N content in the soil (Nmin) was measured according to the standard protocol from the Association of German Agricultural Analytic and Research Institutes (VDLUFA), where 20 g of fresh soil was suspended with 200 mL 12.5 mM $CaCl_2$ solution and shaken overhead for 2 h. Soil extracts were analysed for nitrate and ammonium via continuous flow analysis (AutoAnalyzer AA3, Seal Analytical, King's Lynn, UK).

The activity of acid and alkaline phosphatases was measured according to [67], and was expressed as mg p-nitrophenol released from a pre-given p-nitrophenyl phosphate solution in 1 g soil after incubation at 37 °C for 1 h (mg p-nitrophenol $g^{-1}$ $h^{-1}$). The activity of dehydrogenase was determined according to [68] by suspending 1 g soil in 0.8% triphenyltetrazolium chloride solution and incubating it at 37 °C for 24 h. Triphenyltetrazolium chloride in suspension was reduced by soil microorganisms to triphenylformazan (TPF), which was extracted with acetone and measured photometrically at 546 nm (Spekol 11, Carl Zeiss Jena, Jena, Germany). The activity was expressed as mg TPF per g soil released within 1 h (mg TPF $g^{-1}$ $h^{-1}$).

Soil pH was measured in 0.01 M $CaCl_2$ at a soil:solution ratio of 1:2.5 using a pH electrode (pH 1100 L, VWR International, Darmstadt, Germany). The SOM content was determined as the difference in soil weight (air-dried and sieved through 2 mm) before and after incineration (550 °C in muffle furnace). Soil pH and SOM content are shown in Table 3.

### 2.4. Statistical Analysis

All statistical analyses were performed using R version 4.0.4 [69] in the RStudio development environment [70]. Data of all treatments were analysed using two-way ANOVA interspecifically (with fertiliser treatment and plant species as factors) and intraspecifically (with fertiliser treatment, block, and accession as factors). When the effect of the factors was significant ($p < 0.05$), post hoc comparisons were carried out using Duncan's new multiple range test ($\alpha = 0.05$) with function duncan.test from the agricolae package [71]. Normal distribution of residuals and homogeneity of variance were checked in all statistical models using the function shapiro.test and leven.test from the car package [72]. If normality assumptions were not met, data were transformed through the boxcox function from the car package [72].

**Table 3.** Soil organic matter (SOM) and pH (CaCl$_2$) of the soil samples taken after harvest in 2020 and 2021 from the six treatments (no P, TSP, ash, manure, compost, and compost + TSP).

| Treatment | 2020 | | | | | | | |
|---|---|---|---|---|---|---|---|---|
| | pH | | | | SOM (%) | | | |
| | Alfalfa | | Red Clover | | Alfalfa | | Red clover | |
| No P | 5.67 ± 0.31 | B | 5.74 ± 0.35 | C | 2.21 ± 0.17 | B | 2.16 ± 0.14 | B |
| TSP | 5.63 ± 0.22 | B | 5.66 ± 0.22 | D | 2.21 ± 0.11 | B | 2.17 ± 0.15 | B |
| Ash | 5.92 ± 0.26 | A | 5.93 ± 0.29 | AB | 2.16 ± 0.06 | B | 2.15 ± 0.09 | B |
| Manure | 5.88 ± 0.21 | A | 5.94 ± 0.25 | AB | 2.47 ± 0.16 | B | 2.51 ± 0.17 | A |
| Compost | 5.78 ± 0.15 | AB | 5.87 ± 0.27 | B | 2.80 ± 0.23 | A | 2.73 ± 0.25 | A |
| Compost + TSP | 5.86 ± 0.22 | A | 5.95 ± 0.27 | A | 2.92 ± 0.33 | A | 2.70 ± 0.10 | A |
| Average | 5.79 ± 0.25 | | 5.85 ± 0.29 | | 2.46 ± 0.35 | | 2.40 ± 0.30 | |
| Treatment | 2021 | | | | | | | |
| | pH | | | | SOM (%) | | | |
| | Alfalfa | | Red Clover | | Alfalfa | | Red Clover | |
| No P | 5.56 ± 0.30 | BC | 5.51 ± 0.26 | B | 2.00 ± 0.25 | C | 2.02 ± 0.23 | C |
| TSP | 5.52 ± 0.26 | C | 5.52 ± 0.24 | B | 2.05 ± 0.16 | C | 2.01 ± 0.08 | C |
| Ash | 5.76 ± 0.33 | A | 5.72 ± 0.32 | A | 2.00 ± 0.17 | C | 1.88 ± 0.20 | C |
| Manure | 5.73 ± 0.25 | A | 5.70 ± 0.22 | A | 2.28 ± 0.18 | B | 2.23 ± 0.14 | B |
| Compost | 5.69 ± 0.18 | AB | 5.71 ± 0.17 | A | 2.57 ± 0.24 | A | 2.45 ± 0.11 | A |
| Compost + TSP | 5.70 ± 0.27 | AB | 5.73 ± 0.23 | A | 2.60 ± 0.23 | A | 2.53 ± 0.19 | A |
| Average | 5.66 ± 0.28 | | 5.65 ± 0.26 | | 2.25 ± 0.32 | | 2.18 ± 0.29 | |

No P = without P supply, TSP = triple-super-phosphate, ash = biomass ash. Manure = cattle manure, compost = biowaste compost. Values are means of four block replicates ± standard deviation. Letters in capital case indicate a significant difference between treatments (Duncan's new multiple range test with $p < 0.05$).

## 3. Results

### 3.1. Plant Yield and Nutrient Uptake

The crop biomass differed mainly in dependence on applied fertilisers and cultivation years, while the differences between alfalfa and red clover were relatively small (4.7 vs. 4.9 Mg ha$^{-1}$ in 2020 and 12.0 vs. 10.5 Mg ha$^{-1}$ in 2021, $p < 0.05$). Within each plant species, a similar pattern of intraspecific differences was found in both years (Figure 2). The biomass of most alfalfa accessions was measured at a close range (4.7 to 5.0 Mg ha$^{-1}$ in 2020 and 11.6 to 12.9 Mg ha$^{-1}$ in 2021) and only accession LE2812 showed a significantly lower value than the others (3.8 and 8.4 Mg ha$^{-1}$ in 2020 and 2021, respectively). More pronounced intraspecific diversity was found between red clover accessions (4.1 to 5.4 Mg ha$^{-1}$ in 2020 and 9.1 to 11.9 Mg ha$^{-1}$ in 2021). Phosphorus supply increased the average biomass of both crop species in comparison to the no P treatment in 2020 and followed the order compost ≥ compost + TSP > manure > TSP ≥ ash ($p < 0.05$). However, the difference between treatments in 2021 was less pronounced, and only in the ash and no P treatment lower biomasses were found than in the others ($p < 0.05$). The combined treatment of compost + TSP did not further increase crop biomass compared with single compost application in any year.

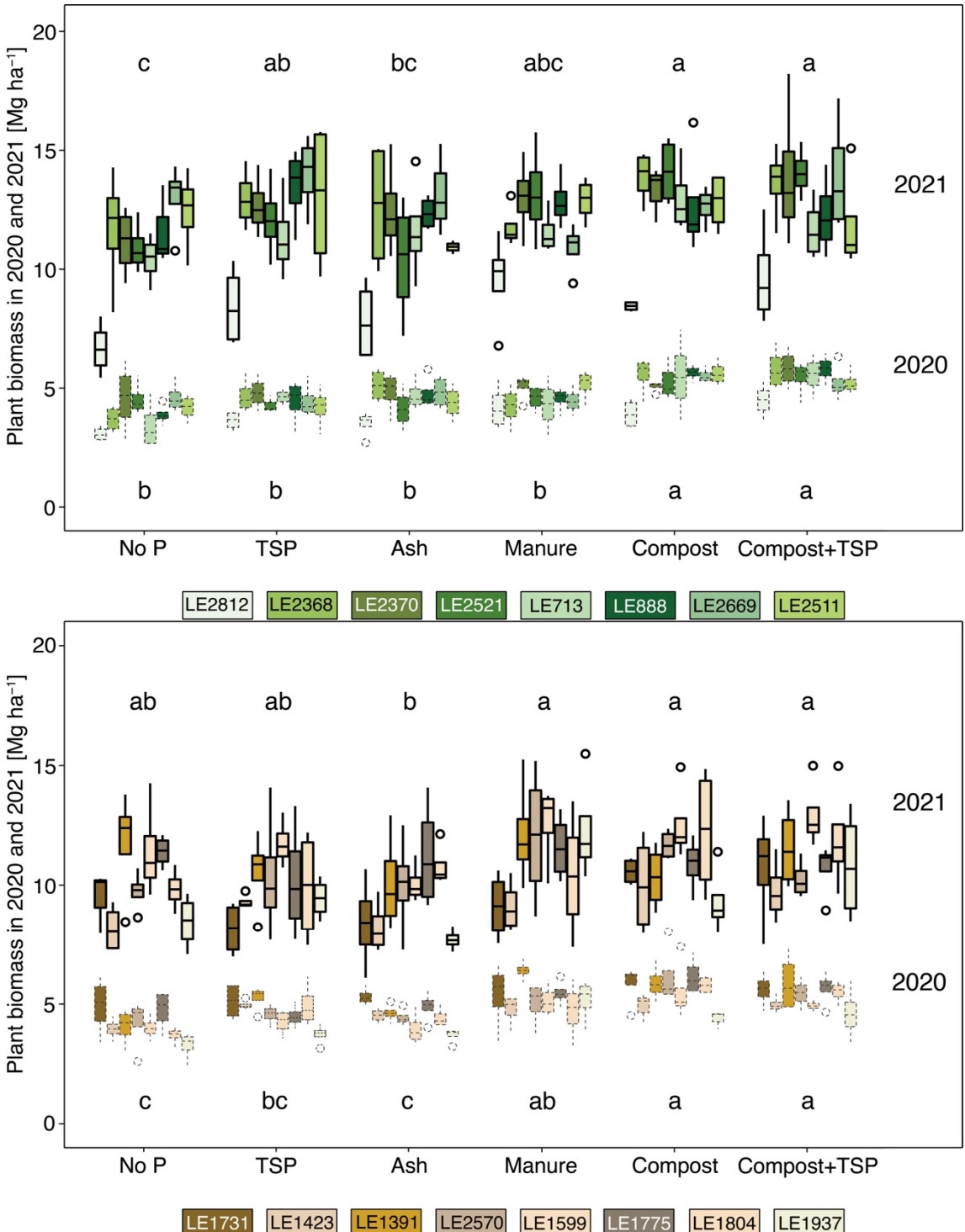

**Figure 2.** Plant biomass of selected accessions of alfalfa (upper part) and red clover (lower part) of the six studied treatments (no P = without P supply, TSP = triple-super-phosphate, ash = biomass ash, Manure = cattle manure, compost = biowaste compost, and compost + TSP) in 2020 and 2021. Alfalfa and red clover accessions are illustrated in different shades of green and brown colours, respectively. The years 2020 and 2021 are differentiated using dotted and solid lines, respectively. Letters indicate a significant difference between treatments (based on the average of accessions) within the same plant species (Duncan's new multiple range test with $p < 0.05$). Mean ± standard deviation (n = 4).

The average plant P concentration of red clover (2.76 g kg$^{-1}$) was slightly but significantly higher than that of alfalfa (2.62 g kg$^{-1}$) in 2020 (Table 4), but decreased to the same range in 2021 (2.55 vs. 2.53 g kg$^{-1}$) (Table 5). Significant, though low, intraspecific differences were only detected in red clover. Combined P application with compost + TSP resulted in higher P concentrations than the no P treatment (2.82 vs. 2.55 g kg$^{-1}$ in 2020 and 2.66 vs. 2.45 g kg$^{-1}$ in 2021). No interactive effects were measured between plant species (or accessions) and fertiliser treatments.

Plant P uptake was strongly correlated to the crop biomass (Pearson's correlation coefficient r = 0.908 in 2020 and 0.827 in 2021, $p < 0.001$) and, consequently, alfalfa showed a lower average P uptake in 2020 compared to red clover but a higher average P uptake in 2021 (Figure 3). The alfalfa accession LE2812 had a significantly lower P uptake (9.6 and 21.5 kg ha$^{-1}$ in 2020 and 2021, respectively) than other alfalfa accessions ranging from 12.2 to 13.1 in 2020 and 29.9 to 33.5 kg ha$^{-1}$ in 2021. As described for biomass, the P uptake of clover accessions had more intraspecific differences than alfalfa ranging from 12.2 to 14.9 kg ha$^{-1}$ in 2020 and from 24.0 to 28.3 kg ha$^{-1}$ in 2021. The red clover accessions showed the same P uptake pattern in both years. Similarly to biomass, the P uptake of both crops followed the order compost + TSP ≥ compost > manure > TSP ≥ ash > no P in 2020, but the difference was less pronounced between treatments in 2021.

Plant N concentration and uptake were only measured in 2021 (Table 6). The average plant N concentration of alfalfa showed slightly but significantly higher values than red clover (27.6 > 26.8 g kg$^{-1}$). The fertiliser applied, even organic, did not rise the N concentration of either species. Significant intraspecific differences were measured for red clover, with accession LE1423 showing the highest N concentration of 29.5 g kg$^{-1}$. Like with plant P uptake, plant N uptake also closely correlated with plant biomass (r = 0.836, $p < 0.001$) (Figure 4), with alfalfa showing a higher average N uptake than red clover (330 and 277 kg ha$^{-1}$, respectively) ($p < 0.05$). Intraspecific differences were only found between alfalfa accessions, with the lowest values in accession LE2812 (231 kg ha$^{-1}$) in comparison to the other accessions ranging from 320 to 361 kg ha$^{-1}$.

**Table 4.** Phosphorus (P) concentration of selected accessions of alfalfa and red clover from the six studied treatments (no P, TSP, ash, manure, compost, and compost + TSP) in 2020.

| P Concentration (g kg$^{-1}$) | | Treatment | | | | | | | | | | | |
|---|---|---|---|---|---|---|---|---|---|---|---|---|---|
| Species | Accession | No P | | TSP | | Ash | | Manure | | Compost | | Compost + TSP | |
| Alfalfa | LE2812 | 2.42 ± 0.21 | Bb | 2.49 ± 0.16 | ABab | 2.81 ± 0.23 | Aa | 2.55 ± 0.06 | ABa | 2.53 ± 0.16 | ABa | 2.48 ± 0.24 | ABb |
| | LE2368 | 2.41 ± 0.08 | Cb | 2.48 ± 0.12 | BCab | 2.58 ± 0.13 | ABCab | 2.82 ± 0.29 | ABa | 2.72 ± 0.31 | ABCa | 2.89 ± 0.36 | Aa |
| | LE2370 | 2.54 ± 0.07 | Aab | 2.52 ± 0.15 | Aab | 2.57 ± 0.29 | Aab | 2.69 ± 0.30 | Aa | 2.63 ± 0.30 | Aa | 2.60 ± 0.18 | Aab |
| | LE2521 | 2.53 ± 0.25 | Aab | 2.78 ± 0.17 | Aa | 2.52 ± 0.18 | Aab | 2.75 ± 0.26 | Aa | 2.76 ± 0.15 | Aa | 2.83 ± 0.28 | Aa |
| | LE713 | 2.78 ± 0.26 | Aa | 2.39 ± 0.23 | Bb | 2.37 ± 0.20 | Bb | 2.88 ± 0.21 | Aa | 2.69 ± 0.31 | ABa | 2.89 ± 0.18 | Aa |
| | LE888 | 2.62 ± 0.20 | Aab | 2.62 ± 0.37 | Aab | 2.79 ± 0.23 | Aab | 2.67 ± 0.33 | Aa | 2.75 ± 0.37 | Aa | 2.75 ± 0.26 | Aab |
| | LE2669 | 2.37 ± 0.26 | Ab | 2.40 ± 0.11 | Ab | 2.41 ± 0.30 | Aab | 2.60 ± 0.40 | Aa | 2.55 ± 0.29 | Aa | 2.78 ± 0.22 | Aab |
| | LE2521 | 2.43 ± 0.10 | Bb | 2.69 ± 0.13 | ABab | 2.50 ± 0.31 | ABab | 2.56 ± 0.12 | Aab | 2.83 ± 0.31 | Aa | 2.73 ± 0.17 | ABab |
| | Average | 2.51 ± 0.21 | C | 2.54 ± 0.21 | BC | 2.57 ± 0.26 | BC | 2.69 ± 0.26 | AB | 2.68 ± 0.27 | AB | 2.74 ± 0.25 | A |
| Red clover | LE1731 | 2.64 ± 0.03 | Ba | 2.76 ± 0.33 | ABab | 2.71 ± 0.16 | Bab | 2.84 ± 0.23 | ABa | 3.01 ± 0.13 | Aab | 2.84 ± 0.25 | ABbc |
| | LE1423 | 2.73 ± 0.11 | Aa | 2.81 ± 0.39 | Aab | 2.89 ± 0.30 | Aa | 2.83 ± 0.08 | Aa | 3.06 ± 0.13 | Aab | 2.97 ± 0.18 | Ab |
| | LE1391 | 2.36 ± 0.09 | Bb | 2.65 ± 0.15 | Ab | 2.59 ± 0.13 | ABb | 2.78 ± 0.26 | Aa | 2.76 ± 0.15 | Ab | 2.85 ± 0.25 | Abc |
| | LE2750 | 2.54 ± 0.10 | Cab | 2.65 ± 0.16 | BCb | 2.62 ± 0.13 | BCab | 2.72 ± 0.11 | ABa | 2.84 ± 0.04 | Aab | 2.82 ± 0.15 | Abc |
| | LE1599 | 2.55 ± 0.14 | Bab | 2.70 ± 0.21 | ABb | 2.78 ± 0.18 | ABab | 2.70 ± 0.05 | ABa | 2.84 ± 0.35 | ABab | 2.89 ± 0.19 | Abc |
| | LE1775 | 2.58 ± 0.11 | Aab | 2.74 ± 0.12 | Ab | 2.71 ± 0.20 | Aab | 2.63 ± 0.09 | Aa | 2.80 ± 0.20 | Aab | 2.74 ± 0.15 | Ac |
| | LE1804 | 2.55 ± 0.28 | Bab | 2.64 ± 0.17 | ABb | 2.68 ± 0.17 | ABab | 2.71 ± 0.10 | ABa | 2.82 ± 0.23 | ABab | 2.88 ± 0.13 | Abc |
| | LE1937 | 2.79 ± 0.32 | Ba | 3.02 ± 0.45 | ABa | 2.80 ± 0.21 | Bab | 2.86 ± 0.19 | Ba | 3.08 ± 0.23 | ABa | 3.28 ± 0.19 | Aa |
| | Average | 2.59 ± 0.20 | B | 2.75 ± 0.27 | AB | 2.72 ± 0.19 | AB | 2.76 ± 0.15 | AB | 2.89 ± 0.21 | A | 2.90 ± 0.22 | A |
| Average | | 2.55 ± 0.21 | D | 2.65 ± 0.26 | C | 2.64 ± 0.24 | C | 2.72 ± 0.22 | B | 2.78 ± 0.26 | AB | 2.82 ± 0.25 | A |

No P = without P supply, TSP = triple-super-phosphate, ash = biomass ash. Manure = cattle manure, compost = biowaste compost. Letters in capital case indicate a significant difference between treatments and letters in lower case indicate a significant difference between accessions within the same plant species (Duncan's new multiple range test with $p < 0.05$). Mean ± standard deviation (n = 4).

**Table 5.** Phosphorus (P) concentration of selected accessions of alfalfa and red clover from the six studied treatments (no P, TSP, ash, manure, compost, and compost + TSP) in 2021.

| P Concentration (g kg$^{-1}$) | | Treatment | | | | | | | | | | | |
|---|---|---|---|---|---|---|---|---|---|---|---|---|---|
| Species | Accession | No P | | TSP | | Ash | | Manure | | Compost | | Compost + TSP | |
| Alfalfa | LE2812 | 2.56 ± 0.09 | Aab | 2.60 ± 0.27 | Aa | 2.51 ± 0.29 | Aa | 2.48 ± 0.37 | Aa | 2.68 ± 0.34 | Aa | 2.44 ± 0.18 | Ab |
| | LE2368 | 2.70 ± 0.24 | ABa | 2.51 ± 0.23 | ABa | 2.35 ± 0.19 | Ba | 2.64 ± 0.08 | ABa | 2.80 ± 0.39 | Aa | 2.78 ± 0.33 | Aab |
| | LE2370 | 2.44 ± 0.12 | Aabc | 2.48 ± 0.30 | Aa | 2.42 ± 0.12 | Aa | 2.44 ± 0.24 | Aa | 2.40 ± 0.23 | Aa | 2.49 ± 0.21 | Ab |
| | LE2521 | 2.49 ± 0.13 | Aab | 2.51 ± 0.45 | Aa | 2.44 ± 0.13 | Aa | 2.60 ± 0.23 | Aa | 2.58 ± 0.28 | Aa | 2.74 ± 0.35 | Aab |
| | LE713 | 2.42 ± 0.30 | Aabc | 2.53 ± 0.41 | Aa | 2.65 ± 0.21 | Aa | 2.43 ± 0.16 | Aa | 2.76 ± 0.15 | Aa | 2.65 ± 0.18 | Aab |
| | LE888 | 2.57 ± 0.16 | ABab | 2.32 ± 0.26 | Ba | 2.45 ± 0.20 | Ba | 2.60 ± 0.16 | ABa | 2.62 ± 0.20 | ABa | 2.91 ± 0.22 | Aa |
| | LE2669 | 2.17 ± 0.08 | Cc | 2.39 ± 0.07 | BCa | 2.63 ± 0.14 | ABa | 2.52 ± 0.20 | ABCa | 2.84 ± 0.33 | Aa | 2.74 ± 0.29 | ABab |
| | LE2521 | 2.28 ± 0.25 | Abc | 2.47 ± 0.21 | Aa | 2.63 ± 0.19 | Aa | 2.30 ± 0.19 | Aa | 2.69 ± 0.53 | Aa | 2.63 ± 0.30 | Aab |
| | Average | 2.45 ± 0.23 | A | 2.48 ± 0.27 | A | 2.50 ± 0.20 | A | 2.50 ± 0.22 | A | 2.68 ± 0.32 | A | 2.67 ± 0.28 | A |
| Red clover | LE1731 | 2.30 ± 0.35 | Bbc | 2.59 ± 0.07 | ABb | 2.69 ± 0.11 | Aab | 2.61 ± 0.28 | ABa | 2.57 ± 0.35 | ABab | 2.55 ± 0.21 | ABbc |
| | LE1423 | 2.80 ± 0.04 | ABa | 2.96 ± 0.12 | ABa | 2.87 ± 0.17 | ABa | 2.58 ± 0.34 | Ba | 2.82 ± 0.56 | ABab | 3.21 ± 0.21 | Aa |
| | LE1391 | 2.21 ± 0.39 | Ac | 2.25 ± 0.14 | Ac | 2.29 ± 0.41 | Acd | 2.44 ± 0.30 | Aab | 2.61 ± 0.34 | Aab | 2.38 ± 0.08 | Ac |
| | LE2750 | 2.33 ± 0.35 | Aabc | 2.42 ± 0.42 | Abc | 2.19 ± 0.25 | Acd | 2.15 ± 0.13 | Ab | 2.42 ± 0.18 | Ab | 2.57 ± 0.16 | Abc |
| | LE1599 | 2.36 ± 0.34 | Aabc | 2.48 ± 0.18 | Abc | 2.13 ± 0.11 | Ad | 2.48 ± 0.37 | Aab | 2.53 ± 0.36 | Ab | 2.49 ± 0.25 | Abc |
| | LE1775 | 2.41 ± 0.29 | Aabc | 2.33 ± 0.23 | Abc | 2.39 ± 0.39 | Abcd | 2.38 ± 0.25 | Aab | 2.70 ± 0.55 | Aab | 2.62 ± 0.20 | Abc |
| | LE1804 | 2.42 ± 0.40 | Aabc | 2.36 ± 0.03 | Abc | 2.55 ± 0.18 | Aabc | 2.40 ± 0.20 | Aab | 2.45 ± 0.17 | Ab | 2.47 ± 0.38 | Ac |
| | LE1937 | 2.76 ± 0.29 | ABab | 2.51 ± 0.25 | Bbc | 2.78 ± 0.31 | ABa | 2.63 ± 0.40 | ABa | 3.07 ± 0.14 | Aa | 2.94 ± 0.51 | ABab |
| | Average | 2.45 ± 0.35 | A | 2.49 ± 0.28 | A | 2.49 ± 0.35 | A | 2.46 ± 0.30 | A | 2.65 ± 0.38 | A | 2.65 ± 0.36 | A |
| Average | | 2.45 ± 0.29 | B | 2.48 ± 0.27 | B | 2.50 ± 0.29 | B | 2.48 ± 0.26 | B | 2.67 ± 0.35 | A | 2.66 ± 0.32 | A |

No P = without P supply, TSP = triple-super-phosphate, ash = biomass ash. Manure = cattle manure, compost = biowaste compost. Letters in capital case indicate a significant difference between treatments and letters in lower case indicate a significant difference between accessions within the same plant species (Duncan's new multiple range test with $p < 0.05$). Mean ± standard deviation (n = 4).

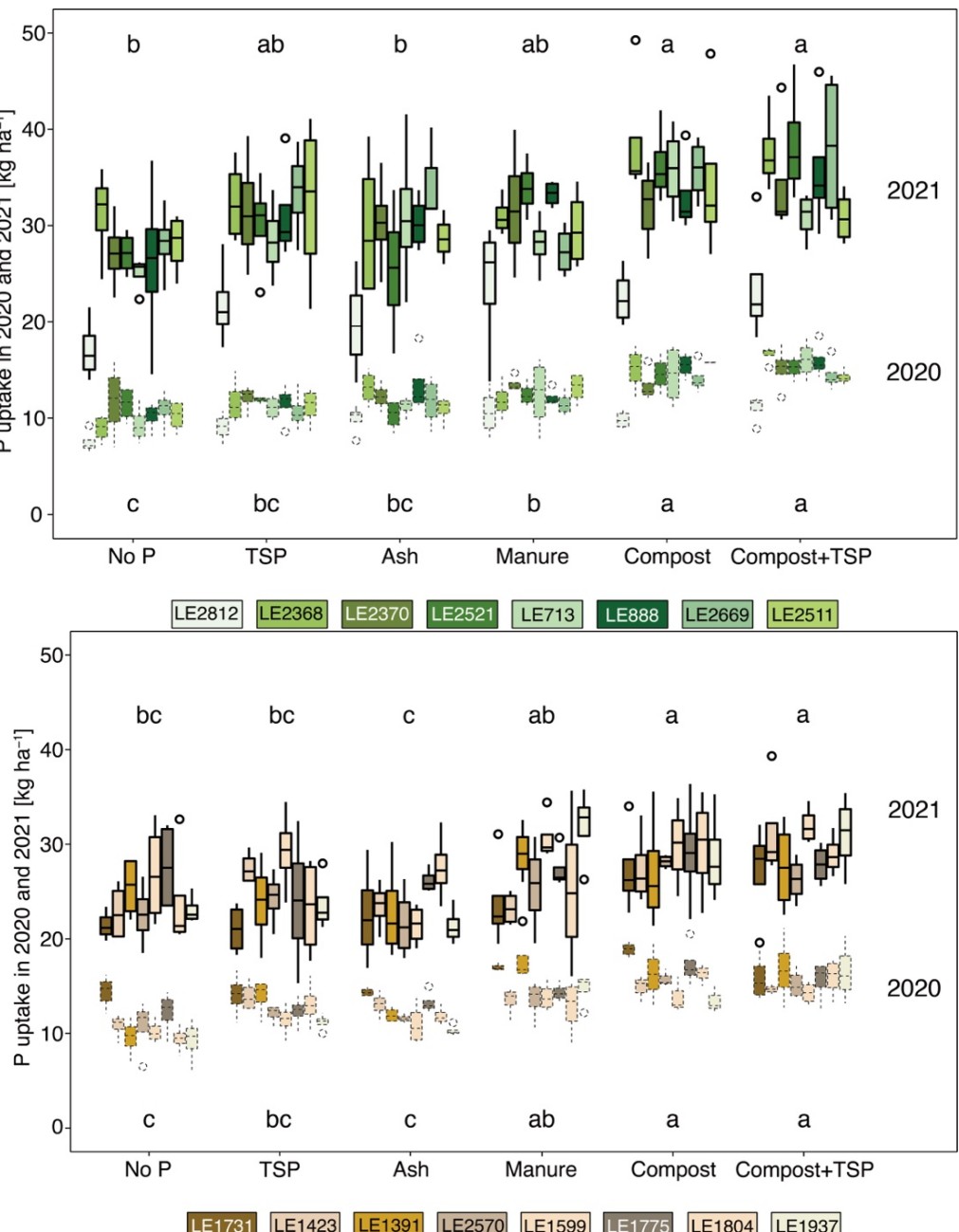

**Figure 3.** Plant phosphorus (P) uptake of selected accessions of alfalfa (upper part) and red clover (lower part) of the six studied treatments (no P, TSP, ash, manure, compost, and compost + TSP) in 2020 and 2021. Alfalfa and red clover accessions are illustrated in different shades of green and brown colours, respectively. The years 2020 and 2021 are differentiated with dotted and solid lines, respectively. Letters indicate a significant difference between treatments (based on the average of accessions) within the same plant species (Duncan's new multiple range test with $p < 0.05$). Mean $\pm$ standard deviation (n = 4).

Table 6. Nitrogen (N) concentration of selected accessions of alfalfa and red clover of the six studied treatments (no P, TSP, ash, manure, compost, and compost + TSP) in 2021.

| N Concentration (g kg⁻¹) | | Treatment | | | | | | | | | | | |
|---|---|---|---|---|---|---|---|---|---|---|---|---|---|
| Species | Accession | No P | | TSP | | Ash | | Manure | | Compost | | Compost + TSP | |
| Alfalfa | LE2812 | 25.5 ± 2.2 | Abc | 27.7 ± 2.0 | Aa | 27.7 ± 2.9 | Aab | 28.1 ± 6.2 | Aa | 28.4 ± 3.4 | Aa | 27.1 ± 2.5 | Aa |
| | LE2368 | 30.3 ± 1.6 | Aa | 28.6 ± 4.8 | Aba | 24.4 ± 2.7 | Bb | 29.8 ± 4.6 | Aa | 28.9 ± 2.5 | Aba | 27.8 ± 3.7 | Aba |
| | LE2370 | 27.1 ± 1.0 | Aabc | 28.0 ± 4.1 | Aa | 29.7 ± 2.4 | Aa | 27.7 ± 3.7 | Aa | 28.7 ± 3.9 | Aa | 28.9 ± 2.9 | Aa |
| | LE2521 | 28.0 ± 2.3 | Aab | 26.3 ± 4.2 | Aa | 26.0 ± 1.8 | Aab | 29.2 ± 3.9 | Aa | 28.9 ± 0.8 | Aa | 28.7 ± 3.1 | Aa |
| | LE713 | 27.4 ± 3.0 | Aabc | 26.6 ± 2.0 | Aa | 29.0 ± 4.2 | Aa | 25.4 ± 2.1 | Aa | 27.5 ± 2.7 | Aa | 29.8 ± 6.7 | Aa |
| | LE888 | 27.1 ± 1.6 | Ababc | 24.4 ± 2.8 | Ba | 28.0 ± 2.1 | Abab | 28.8 ± 1.6 | Aa | 28.6 ± 4.1 | Aba | 26.7 ± 2.3 | Aba |
| | LE2669 | 24.9 ± 3.1 | Abc | 26.3 ± 3.4 | Aa | 30.0 ± 2.5 | Aa | 25.6 ± 2.9 | Aa | 27.1 ± 1.4 | Aa | 25.4 ± 2.3 | Aa |
| | LE2521 | 23.6 ± 3.7 | Ac | 28.1 ± 6.2 | Aa | 29.7 ± 2.7 | Aa | 27.2 ± 3.2 | Aa | 28.1 ± 3.8 | Aa | 27.4 ± 2.8 | Aa |
| | Average | 26.7 ± 2.9 | A | 27.0 ± 3.7 | A | 27.9 ± 3.1 | A | 27.7 ± 3.6 | A | 28.2 ± 2.7 | A | 27.7 ± 3.4 | A |
| Red clover | LE1731 | 27.8 ± 3.3 | Abab | 29.2 ± 2.2 | Aab | 29.9 ± 2.4 | Aab | 24.9 ± 2.3 | Bab | 24.8 ± 1.4 | Bb | 26.8 ± 1.9 | Abab |
| | LE1423 | 32.1 ± 0.9 | Aa | 31.3 ± 1.7 | Aa | 31.6 ± 3.5 | Aa | 25.1 ± 2.5 | Cab | 27.4 ± 1.4 | Bcab | 29.6 ± 2.9 | Aba |
| | LE1391 | 25.8 ± 4.6 | Ab | 25.2 ± 3.9 | Ab | 24.8 ± 3.5 | Ac | 27.5 ± 4.1 | Aa | 27.2 ± 2.9 | Ab | 23.6 ± 3.4 | Abc |
| | LE2750 | 28.0 ± 3.1 | Aab | 26.7 ± 3.7 | Ab | 24.2 ± 2.2 | Abc | 22.3 ± 1.9 | Bb | 25.5 ± 3.0 | Abb | 25.4 ± 0.6 | Abbc |
| | LE1599 | 26.7 ± 2.4 | Aab | 26.6 ± 0.3 | Ab | 24.8 ± 3.1 | Abc | 25.8 ± 1.4 | Aab | 24.7 ± 0.8 | Abb | 22.3 ± 2.3 | Bc |
| | LE1775 | 28.2 ± 3.5 | Aab | 26.7 ± 1.8 | Ab | 26.2 ± 3.1 | Abc | 25.4 ± 2.8 | Aab | 29.0 ± 4.7 | Aab | 26.3 ± 2.7 | Ab |
| | LE1804 | 25.8 ± 4.6 | Ab | 26.4 ± 1.6 | Ab | 27.7 ± 2.5 | Aabc | 26.0 ± 4.1 | Aab | 26.8 ± 2.9 | Ab | 26.1 ± 3.1 | Ab |
| | LE1937 | 27.7 ± 2.2 | Abab | 25.3 ± 1.8 | Bb | 30.4 ± 4.3 | Abab | 27.3 ± 4.2 | Aba | 32.3 ± 5.6 | Aa | 24.7 ± 2.0 | Bbc |
| | Average | 27.8 ± 3.5 | A | 27.2 ± 2.9 | A | 27.5 ± 3.9 | A | 25.6 ± 3.1 | B | 27.3 ± 3.7 | A | 25.6 ± 3.0 | B |
| Average | | 27.2 ± 3.2 | A | 27.1 ± 3.3 | A | 27.7 ± 3.5 | A | 26.6 ± 3.5 | A | 27.8 ± 3.3 | A | 26.7 ± 3.4 | A |

No P = without P supply, TSP = triple-super-phosphate, ash = biomass ash. Manure = cattle manure, compost = biowaste compost. Letters in capital case indicate a significant difference between treatments and letters in lower case indicate a significant difference between accessions within the same plant species (Duncan's new multiple range test with $p < 0.05$). Mean ± standard deviation (n = 4).

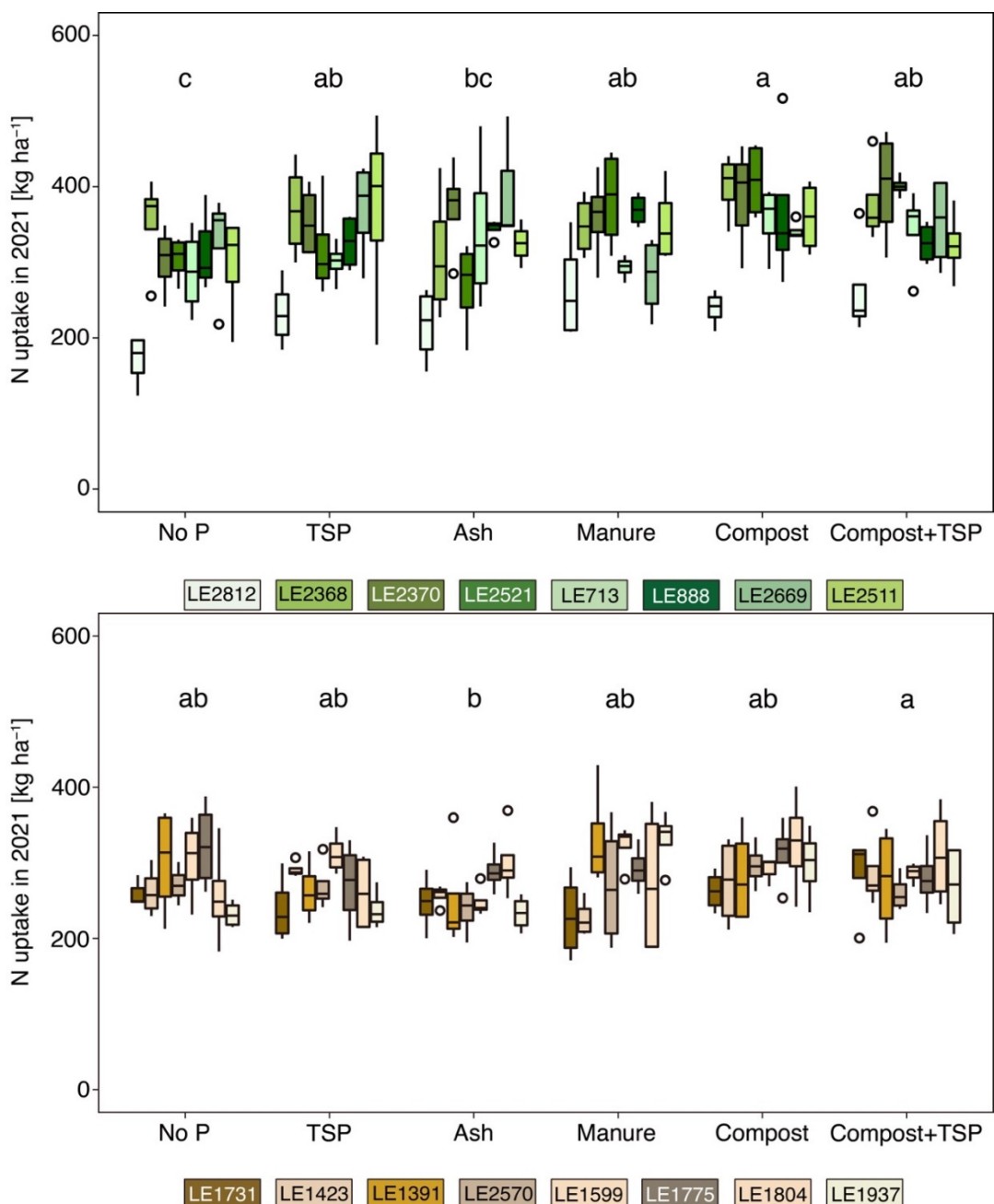

**Figure 4.** Plant nitrogen (N) uptake of selected accessions of alfalfa (upper part) and red clover (lower part) of the six studied treatments (no P, TSP, ash, manure, compost, and compost + TSP) in 2021. Alfalfa and red clover accessions are illustrated in different shades of green and brown colours, respectively. Letters indicate a significant difference between treatments (based on the average of accessions) within the same plant species (Duncan's new multiple range test with $p < 0.05$). Mean $\pm$ standard deviation (n = 4).

The average N:P ratios of alfalfa and red clover were similar (10.9 and 10.7, respectively) in 2021. The fertilisers applied had no effect on the N:P ratio of alfalfa, whereas the combined treatment of compost + TSP reduced the N:P ratio of red clover in comparison to the no P treatment (9.7 < 11.4, $p < 0.05$). No intraspecific differences in the N:P ratio were detected between the alfalfa and red clover accessions.

### 3.2. Nutrient Concentration and Enzyme Activity in the Topsoil

The plant available nutrients (Pdl and Nmin) and enzyme activities in the topsoil were affected by cultivated crops, applied fertilisers, and cultivation year.

A negative correlation was found between plant P uptake and Pdl with correlation factors of about r = −0.3. However, surprisingly, despite the higher P uptake in red clover (see Section 3.1), soil cultivated with red clover showed higher average soil Pdl contents (48.1 mg kg$^{-1}$) than that with alfalfa (41.4 mg kg$^{-1}$) in 2020 ($p < 0.05$), with the most pronounced difference in the combined treatment of compost + TSP (Figure 5). In 2021, the soil Pdl contents were generally lower than in 2020, with only small differences in dependence of the crops cultivated (36.9 and 35.4 mg kg$^{-1}$ for alfalfa and red clover, respectively). The soil Pdl contents were not affected by the individual accessions within a crop species. The Pdl contents in soil strongly depended on applied fertilisers and followed the same order in both cultivation years: compost + TSP > compost > manure > ash > TSP > no P.

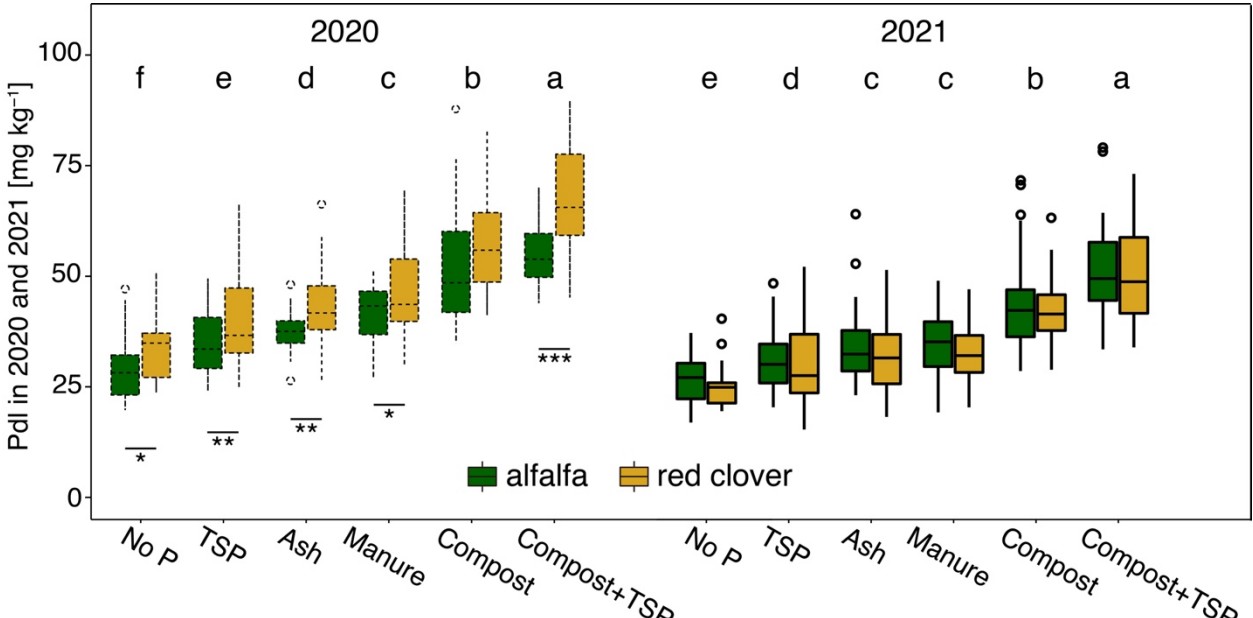

**Figure 5.** Double lactate extractable phosphorus (Pdl) in the topsoil (0 to 30 cm) of the six studied treatments (no P, TSP, ash, manure, compost, and compost + TSP) with cultivation of alfalfa (green) and red clover (brown) in 2020 (framed with dotted lines) and 2021 (framed with solid lines). Letters indicate a significant difference between treatments (based on average of plant species) within the same year (Duncan's new multiple range test with $p < 0.05$). Whiskers and asterisks indicate a significant interspecific difference between the respective treatments. Mean ± standard deviation (n = 32).

Soil cultivated with alfalfa had higher average Nmin contents in soil in 2020 and 2021 (18.2 and 16.4 kg ha$^{-1}$, respectively) than red clover (16.2 and 13.9 kg ha$^{-1}$, respectively) (Figure 6). Intraspecific differences were not detected. The organic fertilisers compost and manure increased soil Nmin content in comparison to treatments without additional N amendment slightly (but significantly), with 17.8, 18.3, and 19.7 kg ha$^{-1}$ measured for manure, compost, and compost + TSP, respectively, and 13.1, 13.5, and 15.1 kg ha$^{-1}$ measured for no P, TSP, and ash, respectively. The Nmin content in the topsoil decreased from 2020 to 2021 with a more pronounced reduction found for red clover (−11.5% in 2021) ($p < 0.05$).

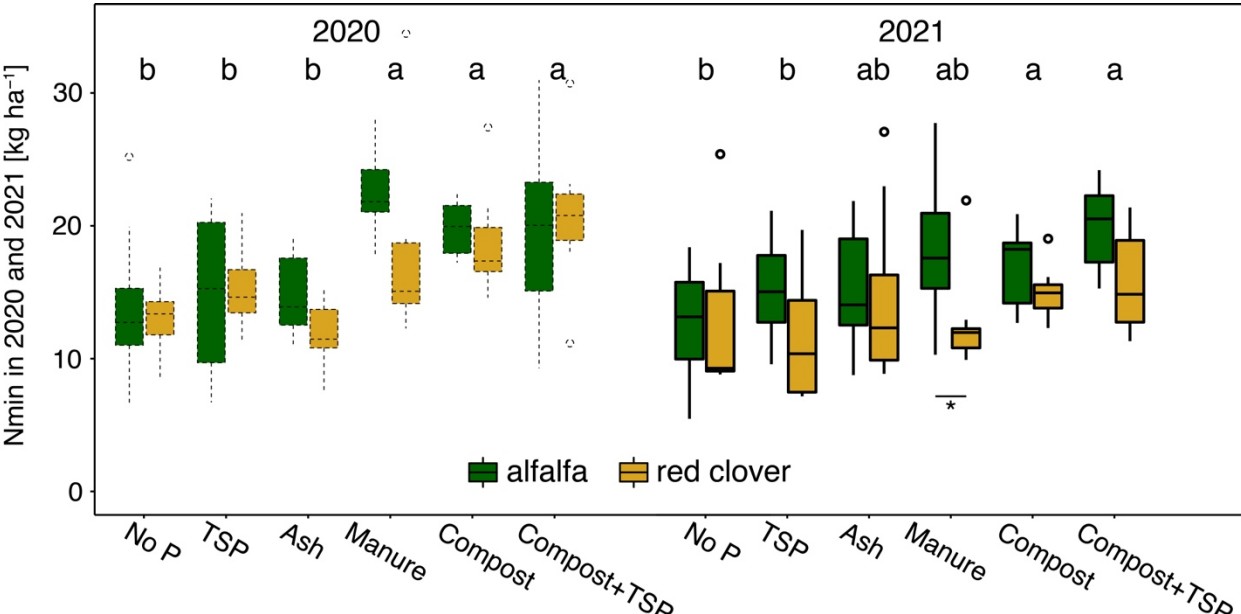

**Figure 6.** Mineral nitrogen (Nmin) content in the topsoil (0 to 30 cm) of the six studied treatments (no P, TSP, ash, manure, compost, and compost + TSP) with cultivation of alfalfa (green) and red clover (brown) in 2020 (framed with dotted lines) and 2021 (framed with solid lines). Letters indicate a significant difference between treatments (based on average of plant species) within the same year (Duncan's new multiple range test with *p* < 0.05). Whiskers and asterisks indicate a significant interspecific difference between the respective treatments. Mean ± standard deviation (n = 8).

The crops cultivated affected the activities of acP in 2021. The cultivation of red clover resulted in higher values than of alfalfa (112.7 and 107.9 µg p-Nitrophenol g DM$^{-1}$ h$^{-1}$, respectively) (*p* < 0.05). The accessions had relatively small effects on enzyme activities in soil. The highest acP and alP activities (137.6 and 44.8 µg p-Nitrophenol g DM$^{-1}$ h$^{-1}$) were found for red clover accession LE1423 (Supplementary Tables S1 and S2) in 2020. While the crops mainly affected the activity of acP, the fertiliser treatments mainly affected the activities of DH and alP with higher activities found after the application of manure and compost in comparison to the no P treatment (*p* < 0.05) (Figure 7). The combined treatment of compost + TSP did not further increase the enzyme activities compared to compost alone. Although there were some interactions between the crop species and treatments, no clear pattern could be identified. Between two cultivation years, the activities of alP and DH in the topsoil were at the same level, whereas acP activity decreased from 131.4 µg p-Nitrophenol g DM$^{-1}$ h$^{-1}$ in 2020 to 110.3 µg p-Nitrophenol g DM$^{-1}$ h$^{-1}$ in 2021 (average of all crops and fertiliser treatments).

### 3.3. Nutrient Concentration in the Subsoil

The plant available nutrients (Pdl and Nmin) in lower soil depths (30 to 60 and 60 to 90 cm) were measured for selected plots, considering only two accessions of each species (see Section 2.2). The average Pdl content of both years decreased with increasing soil depths from 40.1 mg kg$^{-1}$ in 0 to 30 cm to 28.1 mg kg$^{-1}$ in 30 to 60 cm and 7.5 mg kg$^{-1}$ in 60 to 90 cm soil depth (Figure 8). Unlike in the topsoil, neither interspecific nor intraspecific differences in the subsoil were detected regarding the Pdl content. A clear fertiliser effect on Pdl was found in a depth of 30 to 60 cm. Here, compost and manure application increased the Pdl contents from 20.7 mg kg$^{-1}$ in the no P treatment to 34.3 and 31.2 mg kg$^{-1}$, respectively (*p* < 0.05) (average of crop species and cultivation years). In the soil depth of 60 to 90 cm, however, the fertilisers did not affect the Pdl contents.

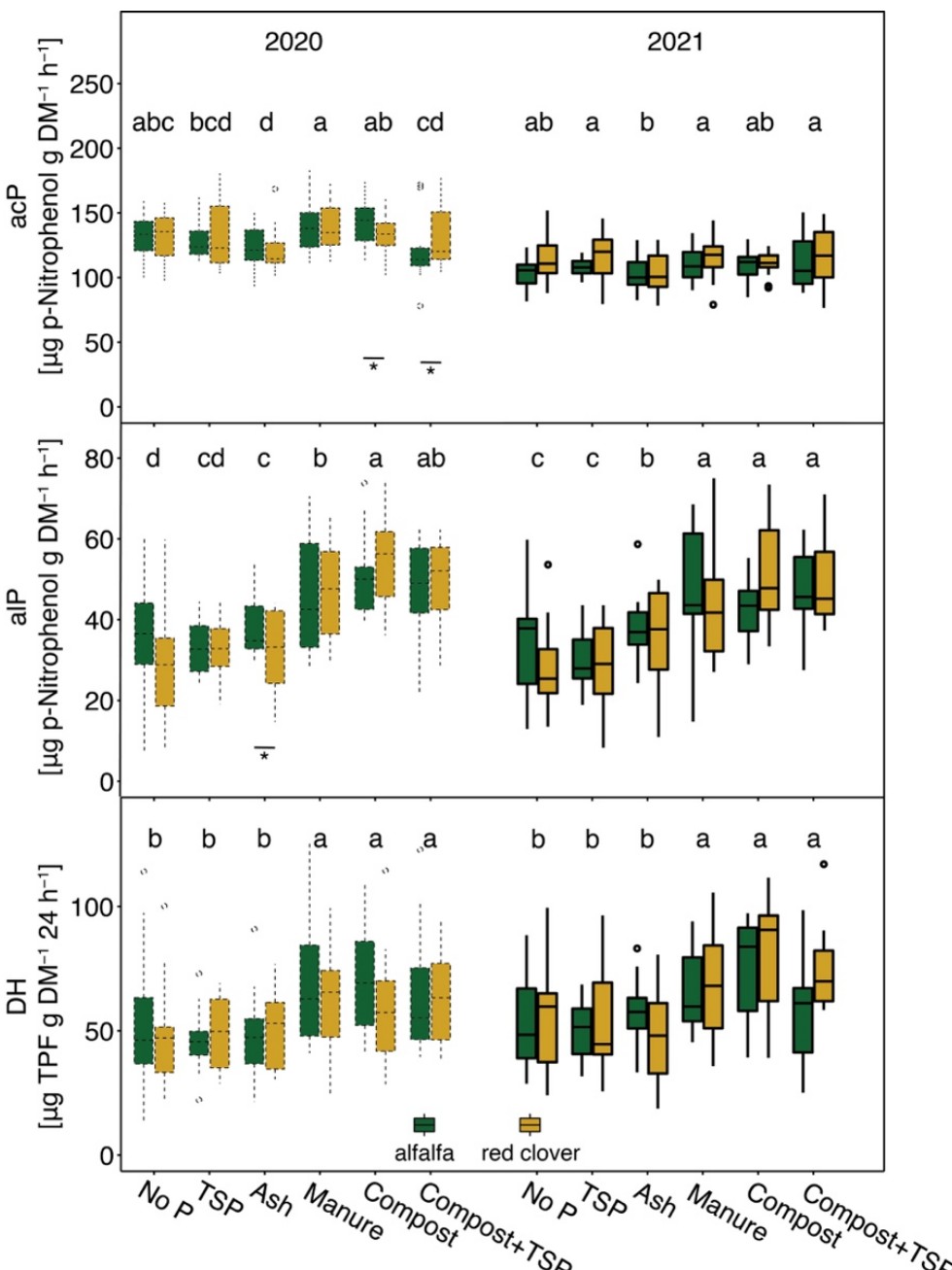

**Figure 7.** Activities of acid and alkaline phosphatase (acP and alP) and dehydrogenase (DH) in 0 to 20 cm soil depth of the six studied treatments (no P, TSP, ash, manure, compost, and compost + TSP) with cultivation of alfalfa (green) and red clover (brown) in autumn of 2020 (framed with dotted lines) and 2021 (framed with solid lines). Letters indicate a significant difference between treatments (based on average of plant species) within the same year (Duncan's new multiple range test with $p < 0.05$). Whiskers and asterisks indicate a significant interspecific difference between the respective treatments. Mean ± standard deviation (n = 12).

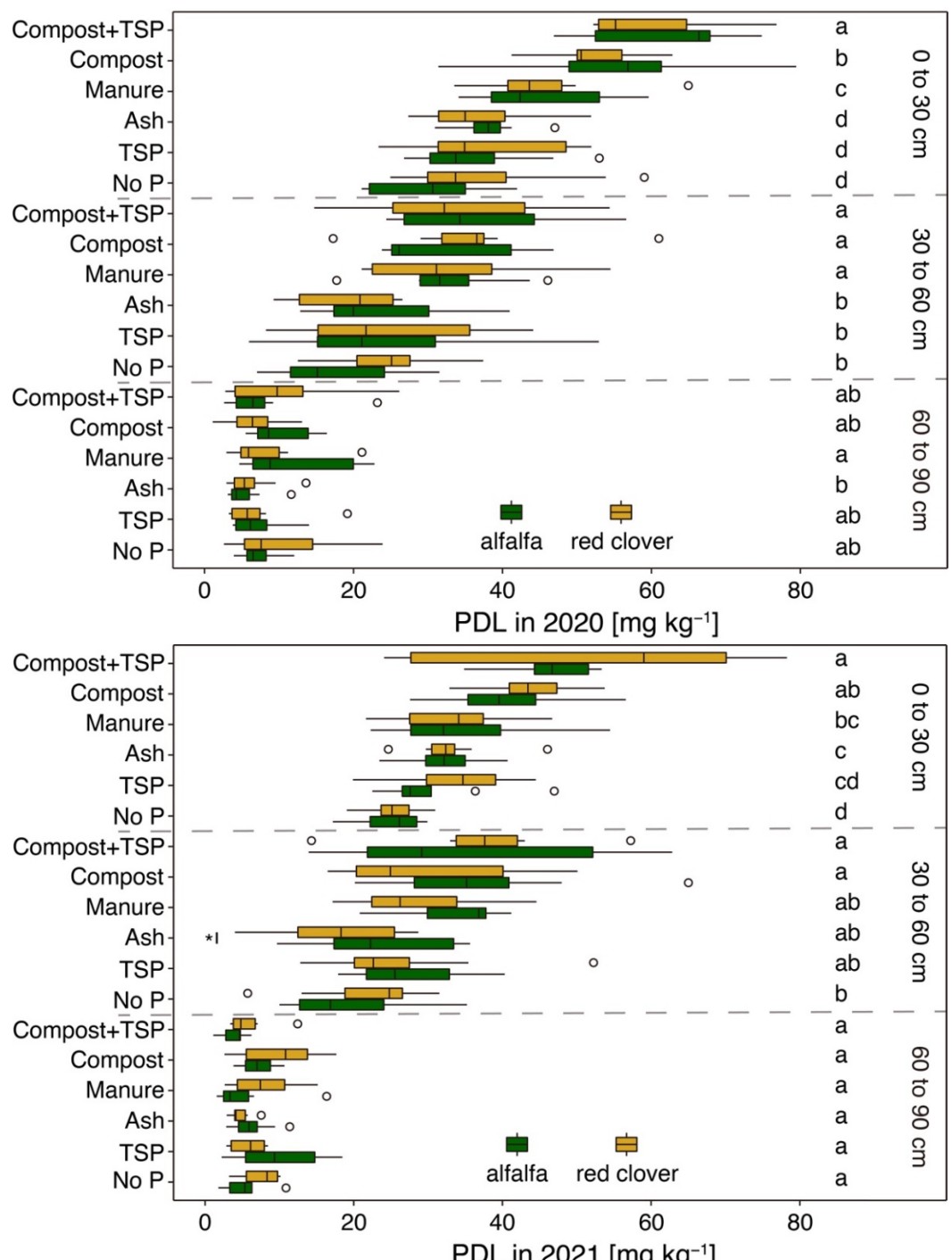

**Figure 8.** Double lactate extractable phosphorus (Pdl) content of soil samples taken in autumn of 2020 and 2021 in three soil depths (0 to 30, 30 to 60, and 60 to 90 cm) of the six studied treatments (no P, TSP, ash, manure, compost, and compost + TSP) with cultivation of alfalfa (green) and red clover (brown). Letters indicate a significant difference between treatments (based on average of plant species) within the same depth (Duncan's new multiple range test with $p < 0.05$). Whiskers and asterisks indicate a significant interspecific difference between the respective treatments. Mean $\pm$ standard deviation (n = 8).

The crops clearly affected the Nmin contents in deeper soil. In the alfalfa plots, the Nmin content decreased from 18.2 in the topsoil to 9.4 until 60 cm and to 4.8 kg ha$^{-1}$ until 90 cm (Figure 9). However, for red clover the Nmin contents did not further decrease from

the 30 to 60 to the 60 to 90 cm depth (12.6 and 11.8 kg ha$^{-1}$, respectively). Consequently, clearly higher Nmin contents were found in the red clover plots than in the alfalfa plots in the soil depth of 60 to 90 cm. Differences between the accessions were not measured. Compared with the no P treatment (8.7 kg ha$^{-1}$ based on the average of plant species), the manure and compost treatments raised the Nmin in 30 to 60 cm to 11.5 and 14.7 kg ha$^{-1}$, respectively.

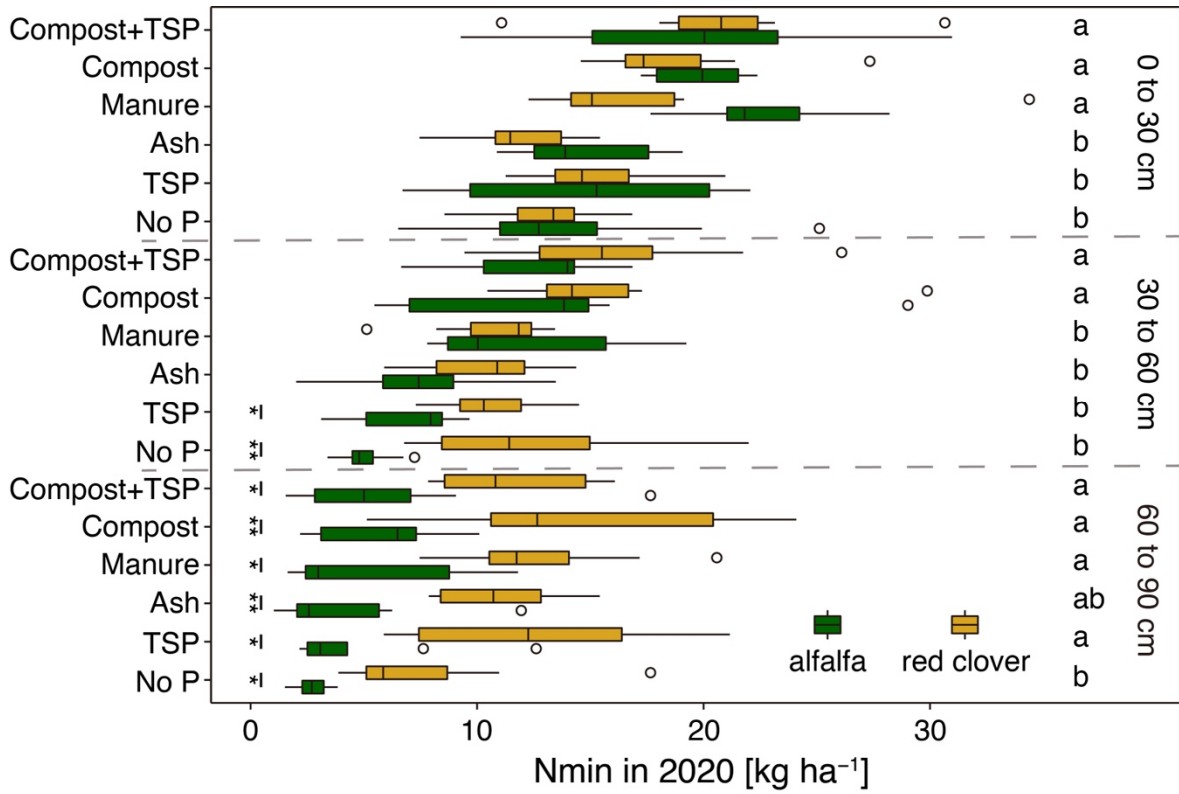

**Figure 9.** Mineral nitrogen (Nmin) content of soil samples taken in autumn 2020 in three soil depths (0 to 30, 30 to 60, and 60 to 90 cm) of the six studied treatments (no P, TSP, ash, manure, compost, and compost + TSP) with cultivation of alfalfa (green) and red clover (brown). Letters indicate a significant difference between treatments (based on average of plant species) within the same depth (Duncan's new multiple range test with $p < 0.05$). Whiskers and asterisks indicate a significant interspecific difference between the respective treatments. Mean ± standard deviation (n = 8).

## 4. Discussion

### 4.1. Inter- and Intraspecific Biomass Production and Nutrient Uptake

Alfalfa and red clover showed significant differences regarding plant biomass and nutrient uptake. However, these were relatively small and also affected by the cultivation year. While red clover had higher biomass production than alfalfa in 2020 (4.9 > 4.7 Mg ha$^{-1}$), it was the other way around in 2021 (10.5 < 12.0 Mg ha$^{-1}$). These interspecific patterns of biomass production were also measured in another study under similar conditions in southwest Sweden [58]. However, with dependence on the management and environmental conditions, red clover can also be superior to alfalfa in subsequent cultivation years, as shown in a humid continental climate in Michigan (USA) [73].

On average, the biomass of both alfalfa and red clover almost doubled in the second cultivation year. Higher biomass production in the second year can be explained by a more developed root system. Since both alfalfa and red clover are perennial plants, the expression of agronomic properties is strongly influenced by the stand age of the crop [74]. Furthermore, weather conditions may also have contributed to the large differences in biomass production in both years. A water deficit in spring 2020 after sowing (Figure 1) could have

limited the plant development and productivity, with yields lower than 5 Mg ha$^{-1}$ in the first cultivation year. In contrast, relatively high precipitation in spring 2021 (April and May) could have benefitted plant growth in 2021. Alfalfa has high drought tolerance due to its capacity to utilise soil moisture in deep soil via its deep root system [75,76], but a water deficit after sowing can limit plant growth, when the root system is just beginning to form. Relevant differences between biomass production in both vegetation periods with the doubling of yields from 4 to 10 Mg ha$^{-1}$ in the seeding year and up to 20 Mg ha$^{-1}$ in the second year were also measured by the authors of [77–79].

As the nutrient concentrations in the plant tissues varied only slightly in our study, the nutrient uptake was mainly dependent on biomass production. Surprisingly, between the accessions no pronounced differences in the nutrient concentration were found, although they varied widely regarding origin, sample status, and maturity group. This is in contrast to differing P concentrations in the plant tissue measured in the preparatory experiment on two other field sites for the selection of accessions (see Section 2.1) and underlines the importance of site effects and management regarding the P accumulation in plants [58,79–81]. In addition, perhaps more pronounced effects would have been found in a pot experiment with limited substrate volumes and thus a more severe P deficiency. Thus, clear differences in the P uptake of twenty-six genotypes of subterranean clover could be found in a pot experiment after five weeks of growth [51]. For alfalfa, accession LE2812 showed significantly lower biomass and nutrient uptake than the other entries, independent of the fertiliser treatment. Accession LE2812 is a traditional cultivar from Yemen which was selected for this experiment because of its high P concentration in biomass. Nevertheless, the environmental conditions in northeast Germany probably limited its plant growth and consequently also P uptake. Intraspecific differences among red clover accessions partly occurred, but without a consistent pattern. They also clearly differed between the cultivation years. For example, red clover accession LE1731, which had the highest average plant biomass and P uptake in the seeding year, only slightly increased its biomass during regrowth in the second year, suggesting it as a suitable candidate for annual rather than perennial cultivation. In contrast, accession LE1599 with the lowest biomass and P uptake in 2020 and great increase in 2021 (biomass plus 156% and P uptake plus 122%) should be instead included in perennial cultivation.

In our study, the average N:P ratio of both crops ranged between 10.1 and 11.2, which was slightly higher than the average N:P ratio of 8.7 in a data-based study by the authors of [82] with 52 legume species. While an additional N supply via the organic fertiliser had almost no effect in our study, the double P supply in the combined fertiliser treatment reduced the N:P ratio in red clover from 11.4 to 9.7 (see Section 3.1). This highlights the role of P uptake as being the main source of variability in N:P stoichiometry, as also concluded by the authors of [82].

### 4.2. Soil Characteristics as Affected by Crops

Crop effects on soil characteristics were found for Pdl in the topsoil (0 to 30 cm) and Nmin in the subsoils (30 to 60 and 60 to 90 cm). Although the topsoil Pdl content was mainly driven by the fertiliser treatments (see Section 4.3), it was also negatively correlated with crop P uptake. This can be explained by the fact that the labile P taken up by the crops could not be replenished from other P pools or fertilisers by the time of sampling shortly after the harvest. Consequently, lower soil Pdl values were found in 2021 when the plants were well established and had a relative high P uptake compared to the seeding year 2020. Deviating from this general relation, red clover resulted in higher soil Pdl contents in 2020 than alfalfa (44.8 vs. 41.4 mg kg$^{-1}$) while both crops had similar P uptakes. This points to active P mobilisation processes in the soil by red clover. In order to evaluate the effects of crops and their associated microbiota on P mobilisation, we considered the phosphatases as enzymes involved in the soil P turnover, as also indicated in previous studies on legumes [83–86]. We found higher activities of acP for red clover than for alfalfa in 2021 but not in 2020. Thus, P mobilisation processes were possibly only partly driven by

enzyme exudation. It has to be considered that in our study the bulk soil was sampled and analysed. As processes of P mobilisation are more pronounced in the rhizosphere, possibly clearer crop effects would have been found in the close vicinity of roots, as shown by the authors of [87,88] in studies with alfalfa.

The Nmin content in 2020 differed in deeper soil depth in dependence of the cultivated crops. Especially in the depth from 60 to 90 cm, alfalfa caused much lower Nmin contents than red clover (4.8 vs. 11.8 kg ha$^{-1}$, $p < 0.05$). This can be only partly explained by a higher N uptake of alfalfa (as measured in 2021). Rather, we assume that the rooting of alfalfa may have contributed to N utilisation from deeper soil layers. Alfalfa develops a root system that differs from that of clover and invests mainly in perennial structures (large taproot), whereas red clover maintains a large share of fine nodal roots with no secondary growth [88]. Reduced nitrate leaching into deeper soil layers in relation to alfalfa cultivation was described for contrasting cropping conditions in, e.g., California [89] and Canada [90], and was also found in intercropping systems [91].

Biological N fixation in living biomasses can prevent N from getting lost [4,92,93]. Both crop species fixed considerable N amounts in the aboveground biomass with ca. 150 and 130 kg ha$^{-1}$ in the second cultivation year for alfalfa and clover, respectively. However, high N content in biomass implies the risk of N leaching after decomposition of the biomass, particularly when ploughed before winter [94–96]. This should be considered when planning crop rotations. As residues of clover were described to decompose more rapidly than those of alfalfa [88], significant N transfers likely occur faster in red clover.

According to our results, the influence of the accessions on soil P and N concentrations and on translocation is almost neglectable. Thus, intraspecific differences between soil characteristics are not expected as long as the accession is adapted to the respective conditions and can develop well.

*4.3. Fertiliser Treatments with Different Agronomic Impacts*

The applied organic fertilisers usually increased both plant biomass and P uptake in comparison to the no P treatment in this experiment. However, P supply with TSP or ash had almost no effect on biomass production, although the soil Pdl content in the control without P supply was classified as low to very low according to German soil P classifications [63,97]. Consequently, higher yields of the legumes due to P application could have been expected, as this was also shown previously for other crops at this site [34]. Clearer impacts of P supply were found in a meta-analysis [98] with suggested P application rates from 26 to 33 kg P ha$^{-1}$ yr$^{-1}$ for alfalfa. Similarly, an annual application rate of 30 kg P ha$^{-1}$ yr$^{-1}$ was found to produce the highest forage yields of red clover [99]. Obviously, in our experiment the P release from less available P pools into the labile P pool, supported by the active P mobilisation by the legumes, was still sufficient to meet the P demand of the crops.

Therefore, the positive effect of organic fertiliser in our study was probably not mainly related to P supply but to other impacts on soil quality, such as soil organic matter, storage of water, and biological activity. Increased soil water holding capacities after the application of organic fertilisers have been measured in several studies, finding either no differences between compost and manure [100] or higher water contents in soils amended with compost compared to manure [101,102]. Water storage becomes increasingly important in this study region as well, mainly due to water shortages in spring or early summer.

The content of soil organic matter was found to be higher after compost than after manure application, which can be explained by the stabilisation of labile organic carbon compounds such as amino acids, peptides, and sugars [103]. In addition, compost can be superior to raw dairy manure in rising soil nutrient levels, as shown in our study and also reported by the authors of [104]. Although the P budgets were almost balanced in all single fertiliser applications (Table 2), higher Pdl contents in the soil depths of 0 to 30 cm and 30 to 60 cm were found for compost. These can be attributed to the activation of biological P turnover processes due to the application of organic matter combined with a

high percentage of mineral P in compost of about 80 to 90% [29]. Higher P application in the combined treatment (compost + TSP) further increased the Pdl content in the topsoil, without having an effect on plant P uptake (see above).

With the organic fertilisers, relevant amounts of N were also supplied (N concentration in manure and compost between 0.7 and 0.9% in the fresh matter). However, the Nmin content in soil showed rather moderate variations in the dependence on the fertilisers with values between 10 and 20 kg ha$^{-1}$ in the topsoil. This indicates that the N applied with the organic fertilisers was mainly incorporated in the soil organic matter and not detectable in the Nmin fraction. In the depth of 60 to 90 cm, no differences between the fertiliser treatment were found and the Nmin content was mainly driven by the crops at this depth (see Section 4.2).

The application of organic fertilisers can stimulate the microbial community [105,106], which was also reflected by the increased activities of alP and DH in soil in our study. As acP is mainly released by plants and not by microbes, it was hardly affected by the fertiliser treatments.

## 5. Conclusions

Although legumes have a relatively high P requirement, the P supply had only small effects on red clover and alfalfa yields in our experiment, indicating high soil P replenishment as well as high active P mobilisation by the legumes. The positive agronomic effects of organic fertilisers, especially biowaste compost, were related to soil quality rather than to P supply. High contents of mineral N in the subsoil can be associated with high N losses into the groundwater. Regarding groundwater protection, alfalfa seems to be more suitable than red clover with lower Nmin contents in a soil depth of 60 to 90 cm. The intraspecific effects were marginal. Thus, variety selection appears to be of little importance with respect to P utilisation in red clover and alfalfa cultivation, as long as the varieties used are adapted to the respective environmental conditions.

**Supplementary Materials:** The following supporting information can be downloaded at: https://www.mdpi.com/article/10.3390/agronomy13030900/s1, Table S1: Activities of acid and alkaline phosphatase (acP and alP) and dehydrogenase (DH) in 0 to 20 cm soil depth of the six studied treatments (no P, TSP, ash, manure, compost and compost+TSP) with cultivation of selected accessions of alfalfa and red clover in autumn 2020.; Table S2: Activities of acid and alkaline phosphatase (acP and alP) and dehydrogenase (DH) in 0 to 20 cm soil depth of the six studied treatments (no P, TSP, ash, manure, compost and compost+TSP) with cultivation of selected accessions of alfalfa and red clover in autumn 2021.

**Author Contributions:** Conceptualisation, B.E.-L., K.J.D. and Y.H.; methodology, all authors; investigation, Y.H. and B.E.-L.; resources, B.E.-L., K.J.D. and E.W.; data curation, B.E.-L. and Y.H.; writing—original draft preparation, Y.H. and B.E.-L.; writing—review and editing, Y.H. and B.E.-L.; visualisation, Y.H.; supervision, B.E.-L. and K.J.D.; funding acquisition, B.E.-L. and K.J.D. All authors have read and agreed to the published version of the manuscript.

**Funding:** This research project is part of the Leibniz ScienceCampus Phosphorus Research Rostock and is (co-)funded by the funding line strategic networks of the Leibniz Association. The APC is funded by Rostock University via the University Library.

**Data Availability Statement:** The data presented in this study are available on request from the corresponding author.

**Acknowledgments:** The investigation of soil characteristics was partly supported by the project InnoSoilPhos (Federal Ministry of Education and Research, No. 031B0509A).

**Conflicts of Interest:** The authors declare no conflict of interest.

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
