# Peer review of "Specific and Intraspecific P Efficiency of Small-Grain Legumes as Affected by Long-Term P Management"

_agronomy, doi:10.3390/agronomy13030900_

Round 1

Reviewer 1 Report

Dear Editor

Thank you very much for your invitation to review this manuscript:

Specific and intraspecific P efficiency of small-grain legumes as affected by long-term P management

Comments

The title should be changed to Specific and intraspecific P efficiency of small-seed legumes as affected by long-term P management.

L15- you should add some details in the abstract

L 30- change grain to seed

L 31,33,34,36- Delete authors name and use numbers instead

L39,41,42,43- Delete authors name and use numbers instead

Delete authors names and add number instead in the all manuscript

You should use small letters in all Figures (like Figurec3)

L 256- You must add another clear Figure

Lines 364-387, Discussion section is very poor and should be improved

In the references section, rewrite the references according to the Journal rules

·         See the comments in the Pdf version

Best regards

Author Response

We would like to sincerely thank the reviewer for the helpful comments. All changes in the manuscript were highlighted.

Comment: The title should be changed to Specific and intraspecific P efficiency of small-seed legumes as affected by long-term P management.

RESPONSE: In the literature mainly small-grain or small-seeded legumes are used. If the reviewer does not mind, we would like to keep the term: small-grain.

Comment: L15- you should add some details in the abstract-

RESPONSE: We added the yield of both species in the experimental years and gave concrete numbers. We also specified the strategies of P mobilization in the first sentence. We have tried to describe all important results in the abstract respecting the limitation of the number of words

Comment: L 30- change grain to seed

RESPONSE: See above - In the literature mainly small-grain or small-seeded legumes are used. If the reviewer does not mind, we would like to keep the term: small-grain.

Comment: L 31,33,34,36- Delete authors name and use numbers instead

Comment: L39,41,42,43- Delete authors name and use numbers instead

RESPONSE: We have given now all references according to the requirements of the journal.

Comment: You should use small letters in all Figures (like Figurec3)

RESPONSE: Usually, we used small letters to indicate differences between the mean values. However, in some table we had to use small as well as capital letters in order to consider both experimental factors. Here, letters in capital case indicate significant difference between treatments and letters in lower case indicate significant difference between accessions

Comment: L 256- You must add another clear Figure

RESPONSE: We inserted a new figure.

Comment: Lines 364-387, Discussion section is very poor and should be improved

RESPONSE: In this section the results are described. We have reworked it to make it clearer to the readers

Comment: In the references section, rewrite the references according to the Journal rules.  See the comments in the Pdf version

RESPONSE: We have now arranged all references according to the journal’s rules.

Reviewer 2 Report

Specific and intraspecific P efficiency of small-grain legumes as affected by long-term P management

This kind of studies is very important and needed as a long term study for management

Very important notice, please

Please follow the instructions of the journal in all MS starting with the system of refs., please!

General comments

1- Please the abstract needs to be informative, more important findings are needed in this section

2- Keywords are absent, please

3- Introduction section, very old and not UpToDate refs., this isn’t acceptable by the publication rules, please, where refs. which published in 2023, 2022, 2021???

Please this section needs to be more precise, please update your refs. and your paragraphs should be on the main topic like 1- on P and its potential in plant nutrition? 2- Legumes and their relation and needing to P? 3- Different possible and suggested approaches to mange P problems in soil? And maybe 4- a comparison between Alfalfa and Red clover

4- Where the analysis of used soil mainly available P content by time??

The authors should add a table including the soil analysis (complete not P only) during the long study period to see the changes in soil properties after cultivation during this long term

5- the used Figures are not clear please, try to use the normal columns with Duncan and SE as well, please

6- Why the authors presented results during 2020 and 2021 although this study is a long -term study for 20 years?

This MS needs minor revision

Author Response

We would like to sincerely thank the reviewer for the helpful comments. All changes in the manuscript were highlighted.

Comment: This kind of studies is very important and needed as a long term study for management. Very important notice, please. Please follow the instructions of the journal in all MS starting with the system of refs., please!

RESPONSE: Thank you very much for the comments and the encouraging statement.  

Comment: Please the abstract needs to be informative, more important findings are needed in this section

RESPONSE: We added the yield of both species in the experimental years and gave concrete numbers. We also specified the strategies of P mobilization in the first sentence. We have tried to describe all important results in the abstract respecting the limitation of the number of words.

Comment: 2- Keywords are absent, please

RESPONSE: We included the following Keywords: organic amendments, alfalfa, red clover, phosphorus utilization

Comment: Introduction section, very old and not UpToDate refs., this isn’t acceptable by the publication rules, please, where refs. which published in 2023, 2022, 2021???

Please this section needs to be more precise, please update your refs. and your paragraphs should be on the main topic like 1- on P and its potential in plant nutrition? 2- Legumes and their relation and needing to P? 3- Different possible and suggested approaches to mange P problems in soil? And maybe 4- a comparison between Alfalfa and Red clover

RESPONSE: Thank you for your suggestion. We mainly followed your advice and arranged the introduction as follows: Role of legumes and biological N fixation; P demand of legumes; P management, P efficiency and P mobilisation of legumes; examples for alfalfa and red clover and intraspecific differences.

We carefully considered the available literature and generally used relevant, frequently cited references. In the reworked version we included further more recent references and deleted some older references.

Comment: Where the analysis of used soil mainly available P content by time??

The authors should add a table including the soil analysis (complete not P only) during the long study period to see the changes in soil properties after cultivation during this long term

RESPONSE; Yes, this study is based on P fertilization strategies. We tried to keep all other nutrients and properties almost constant over the duration of the experiment in order to avoid side effects. Consequently, we balanced nutrients and pH by applying fertilizer and liming amendments. However, differences in soil organic matter occurred during the experimental period, resulting in higher values in the organic fertilizer treatments. We have explained this in the Material and Method section 2.2 – line 140 to 144

Comment: the used Figures are not clear please, try to use the normal columns with Duncan and SE as well, please

RESPONSE: This comment is not entirely clear for us. Instead of using bar plots showing only mean and standard error, we decided to use box plots in order to display the distribution of the biomass/nutrient concentration/nutrient uptake of samples within each treatment with more information (minimum, quartiles, median and maximum). We also provided the results of the Duncan test in the figures, as suggested by the reviewer.

Comment: Why the authors presented results during 2020 and 2021 although this study is a long -term study for 20 years?

RESPONSE: This study with the cultivation of legumes was established on a long-term experiment. The experiment exists since 1998 with different P fertilization treatments and is mainly used to study the agronomic effectiveness of the treatments.  In 2020 and 2021, the legumes were integrated into the experimental design.

Reviewer 3 Report

A study of legume P response using a range of accessions with wide differences in origin on a sandy loam soil in northern Germany. The lack of a P response, despite the study being conducted on soil that would usually be considered responsive according to local soil test P calibrations was unfortunate. It would be useful for the authors to indicate this earlier in the manuscript (the critical lactate-P category) as it was not really noted until the Discussion. Line 128 would be appropriate to indicate that the site was considered marginal for P. I couldn't see any reason for the relative lack of response from the other presented data and have to agree with the authors that the reason behind this will likely remain unknown.

The lack of response should perhaps be moved further forward in the discussion? I was wondering throughout what value there was in the data if the entire study was done at non-responsive P concentrations and this partly explained the introductory comments related to environmental risk - it read like a scramble to find a justification to publish and ultimately petered out through lack of conviction in the discussion... Perhaps a better angle on it concerned the wide range in the accessions and the lack of evidence that breeding and selection have imposed upon tissue P uptake or response? Breeding is often done under non-constrained conditions and therefore selects lines that may be less efficient at acquiring P. Breeding conditions favour lazy root systems. But this study showed little to no evidence of 'laziness' particularly if it was marginal for P and the no-P treatments were relatively unaffected with respect to P acquisition.

I did note the authors seemed unaware of the work of McLachlan et al. on subclover accessions and P responses. It is certainly more relevant than faba bean studies, and is even on Trifolium species, and was done on P responsive soils. The conclusions of their studies support the authors observations that perhaps selecting for P efficiency in clovers and small seeded legumes is not a fruitful path forward - whilst there is variation it is not great enough to exploit usefully.

My suspicion is the study is likely to be cited more if it reflected further on the breeding aspects of the P responses and drew together more fully the observations that variation in P recovery is smaller than expected and selection for intraspecific root traits is a path best avoided?

Author Response

We would like to sincerely thank the reviewer for the helpful comments. We modified the manuscript accordingly. All changes in the manuscript were highlighted.

Comment: A study of legume P response using a range of accessions with wide differences in origin on a sandy loam soil in northern Germany. The lack of a P response, despite the study being conducted on soil that would usually be considered responsive according to local soil test P calibrations was unfortunate. It would be useful for the authors to indicate this earlier in the manuscript (the critical lactate-P category) as it was not really noted until the Discussion. Line 128 would be appropriate to indicate that the site was considered marginal for P. I couldn't see any reason for the relative lack of response from the other presented data and have to agree with the authors that the reason behind this will likely remain unknown.

RESPONSE: Thank you very much for this comment. We have highlighted the non-responsiveness in the summary, results and conclusions and we also added more information in the Material and Method sections and in the Discussion.

In previous studies, including this experimental site, it was shown, that the responsiveness of crops differs. While for maize, potatoes and spring cereals often P fertilizer effects were found, winter cereals almost never had higher yields after P application.

We included this fact into the introduction section, line  58 to 61

According to German soil P test classifications the soil P contents can be considered as low to sufficient considering 40 to 60 mg kg-1 as thresholds to be achieved. For the control the P content is very low. Thus, in principle a yield response could have been expected. We included this fact in the Material and Method section (first paragraph of section 2.2) and reworked the first paragraph of discussion section 4.3. as follows: “However, P supply with TSP or ash had almost no effect on biomass production, although, the soil Pdl content in the control without P supply was classified as low to very low, according to German soil P classifications (Kape, 2019; VDLUFA, 2015). Consequently, higher yields of the legumes due to P application could have been expected as it was also shown previously for other crops at this site (Zicker et al. 2018).

Buczko, U.; van Laak, M.; Bettina Eichler-Löbermann, B.; Gans, W.; Merbach, I.;  Panten, K.; Peiter, E.; Reitz, T.; Spiegel, H.; v. Tucher, S. (2018): Re-evaluation of phosphorus fertilizer recommendations based on meta-analyses of long-term field experiments. Ambio 47, 50 – 61, DOI: 10.1007/s13280-017-0971-1

Zicker, T.; von Tucher, S.; Kavka, M.; Eichler-Löbermann, B. (2018): Soil test phosphorus as affected by phosphorus budgets in two long-term field experiments in Germany. Field Crops Res. 218, 158 – 170. DOI: 10.1016/j.fcr.2018.01.008

Comment: The lack of response should perhaps be moved further forward in the discussion? I was wondering throughout what value there was in the data if the entire study was done at non-responsive P concentrations and this partly explained the introductory comments related to environmental risk - it read like a scramble to find a justification to publish and ultimately petered out through lack of conviction in the discussion... Perhaps a better angle on it concerned the wide range in the accessions and the lack of evidence that breeding and selection have imposed upon tissue P uptake or response? Breeding is often done under non-constrained conditions and therefore selects lines that may be less efficient at acquiring P. Breeding conditions favour lazy root systems. But this study showed little to no evidence of 'laziness' particularly if it was marginal for P and the no-P treatments were relatively unaffected with respect to P acquisition.

RESPONSE: Thank you for this comment. Regarding the value of this study - We think that the low effect of P treatments after so many years is an important result and clearly underlines the role of P mobilization for the tested legumes. On the other hand, the positive effects of organic fertilization showed that soil quality - probably especially water retention - is likely to be significant for legume cultivation. We would have expected that there would be differences between the accessions. However, based on our results we can conclude, that the intraspecific differences in P utilization seem to play little role under these experimental conditions. There were clear differences between the species with respect to N translocation in soil. This fact could be important for water protection areas, for example

Regarding the responsiveness - we have included more details in various sections of the manuscript (see our comments above).

Regarding the role of breeding – we have used not only bred varieties, but also accessions that have never been bred, so we must be cautious in interpreting our results in terms of the impact of breeding. Anyway, as mentioned before, we have included this issue in the abstract and conclusion: “…Thus, variety selection appears to be of little importance with respect to P utilisation in red clover and alfalfa cultivation.”

Comment: I did note the authors seemed unaware of the work of McLachlan et al. on subclover accessions and P responses. It is certainly more relevant than faba bean studies, and is even on Trifolium species, and was done on P responsive soils. The conclusions of their studies support the authors observations that perhaps selecting for P efficiency in clovers and small seeded legumes is not a fruitful path forward - whilst there is variation it is not great enough to exploit usefully.

RESPONSE: Thank you for this advise. We included this study in our manuscript in the introduction. Furthermore, we used this study for our discussion (section 4.1) as follows: “In addition, perhaps more pronounced effects would have been found in a pot experiment with limited substrate volumes and thus more severe P deficiency. Thus, clear differences in P uptake of twenty-six genotypes of subterranean clover could be found in a pot experiment after five weeks of growth (McLachlan et al. 2019)”.

Comment: My suspicion is the study is likely to be cited more if it reflected further on the breeding aspects of the P responses and drew together more fully the observations that variation in P recovery is smaller than expected and selection for intraspecific root traits is a path best avoided?

RESPONSE: Thank you very much for the suggestion. We fully agree with this. We stated in the abstract: “Our results emphasised the high P efficiency of small-gain legumes without pronounced inter- or intraspecific differences”. We further concluded that: “Intraspecific effects were marginal. Thus, variety selection appears to be of little importance with respect to P utilisation in red clover and alfalfa cultivation, as long as the varieties used are adapted to the respective environmental conditions.”